# Two drastically different climate states on an Earth-like terra-planet

Sirisha Kalidindi [1,2], Christian H. Reick[1], Thomas Raddatz[1], Martin Claussen[1,3]

[1]Max Planck Institute for Meteorology, Bundesstraße 53, 20146 Hamburg, Germany

[2]International Max Planck Research School on Earth System Modelling, Bundesstraße 53, 20146 Hamburg, Germany

[3]Center for Earth System Research and Sustainability, Universität Hamburg, Bundesstraße 53, 20146 Hamburg, Germany

*Correspondence to*: Sirisha Kalidindi (sirisha.kalidindi@mpimet.mpg.de)

**Abstract.** We study an Earth-like terra-planet (water limited terrestrial planet) with an overland recycling mechanism bringing fresh water back from the high latitudes to the low latitudes. By performing model simulations for such a planet we find two drastically different climate states for the same set of boundary conditions and parameter values: A Cold and Wet (CW) state with dominant low-latitude precipitation and, a Hot and Dry (HD) state with only high-latitude precipitation. We notice that for perpetual equinox conditions, both climate states are stable below a certain threshold value of background soil albedo while above the threshold only the CW state is stable. Starting from the HD state and increasing background soil albedo above the threshold causes an abrupt shift from the HD state to the CW state resulting in a sudden cooling of about 35°C globally which is of the order of the temperature difference between the present-day and the Snowball Earth state. When albedo starting from the CW state is reduced down to zero the terra-planet does not shift back to the HD state (no closed hysteresis). This is due to the high cloud cover in the CW state hiding the surface from solar irradiation so that surface albedo has only a minor effect on the top of the atmosphere radiation balance. Additional simulations with present day Earth's obliquity all lead to the CW state, suggesting a similar abrupt transition from the HD state to the CW state when increasing obliquity from zero. Our study also has implications for the habitability of Earth-like terra-planets. At the inner edge of the habitable zone, the higher cloud cover in the CW state cools the planet and may prevent the onset of a runaway greenhouse state. At the outer edge, the resupply of water at low latitudes stabilizes the greenhouse effect and keeps the planet in the HD state and may prevent water from getting trapped at high latitudes in frozen form. Overall, the existence of bi-stability in the presence of an overland recycling mechanism hints at the possibility of a wider habitable zone for Earth-like terra-planets at low obliquities.

## 1 Introduction

Recent advancements in observational astrophysics like with the Kepler mission led to the discovery of a vast number of potentially habitable planets (Kopparapu et al. 2014). Habitable planets are planets which can maintain liquid water on their surface (Hart, 1978; Kasting et al., 1993; Kopparapu et al., 2014). The habitability of a planet is influenced by several factors like stellar flux, orbit and planetary properties (Schulze-Makuch et al., 2011). In the case of Earth-like planets, it has been shown that the width of the habitable zone strongly depends on the water cycle and the carbonate-silicate cycle (Kasting et al., 1993; Kopparapu et al., 2013; Zsom et al., 2013). This is because these cycles

5 control the atmospheric concentration of water vapour, carbon dioxide and surface pressure of a planet. In our study, we focus on the role of the water cycle.

   The water cycle on a planet strongly depends on the amount of surface water present which is controlled by a complex interplay between processes like atmospheric circulation, precipitation and cloud formation. Depending on the amount of surface water, habitable planets fall into two classes: aqua-planets – planets covered with global oceans

10 – and terra-planets – planets with vast deserts or vegetated surfaces and a limited amount of water (Dune, Herbert, 1965; Abe et al., 2005, 2011, Leconte et al., 2013). In the present study, we investigate the climate of an Earth-like terra-planet for low obliquities.

   Over the recent years, there has been a growing interest in the study of terra-planets due to three reasons: First, the absence of oceans on a terra-planet helps to isolate the effects of land surface processes and thus aids in

15 better understanding of the land-atmosphere coupling (Aleina et al., 2013; Rochetin et al., 2014; Becker and Stevens, 2015). Second, terra-planets with optically thin atmospheres (like present day Earth's atmosphere) can maintain their inner edge of the habitable zone much closer to their parent star compared to aqua-planets (Zsom et al., 2013). The reason is that limited atmospheric access to water results in a dry climate with water confined to the high latitudes for low obliquities (Abe et al., 2005; Abe et al., 2011; Leconte et al., 2013). In such dry climates, the water vapour

20 feedback is severely muted and the greenhouse warming is substantially lowered which allows dry planets to maintain habitability even at higher stellar fluxes (Zsom et al., 2013). Third, terra-planets at low obliquities can support a wider habitable zone compared to that of aqua-planets (Abe et al., 2011) because the dry atmosphere of terra-planets limits the escape of hydrogen molecules and shows a higher resistance to the runaway greenhouse effect. Also, the dry atmosphere inhibits the formation of clouds, ice and snow and thus helps the planet to resist complete freezing (Abe

25 et al., 2011; Leconte et al., 2013). However, the habitable areas on such a dry planet are confined to the edges/bottom of frozen ice caps (Leconte et al., 2013; Zsom et al., 2013). Whether at such edges liquid water can exist sufficiently permanent for life to evolve and persist is unclear (Zsom et al., 2013). Recycling mechanisms similar to those which occur on the present day Earth like ocean circulation and surface runoff must exist to maintain a long-lasting liquid water inventory. Leconte et al., 2013 argue that on dry planets, mechanisms like gravity driven ice flows and

30 geothermal flux can maintain sufficient amounts of liquid water at the edges/bottom of large ice caps. The liquid water thus formed can eventually flow back to the low latitudes to be re-available for evaporation. However, there exists no climate modelling study on an Earth-like terra-planet which actually implements either implicitly or explicitly such a recycling mechanism bringing fresh water back from high to low latitudes. In our study, we consider an Earth-like terra-planet with an unlimited subsurface water reservoir as a way to mimic the recycling mechanism. Even though

35 the water reservoir in our case is unlimited, it is not similar to an aqua-planet because it includes additional resistances which restrict atmospheric access to water (i.e. soil and plant resistances along with aerodynamic resistance).

   Evidence from paleo-climate modelling studies reveals that planets within the habitable zone can support multiple climate states but all of these states may not satisfy the stable liquid water on their surface. Like our Earth can exist in two different climate states – the present day warm state with abundant surface liquid water and the

40 Snowball Earth state with no surface liquid water (Marotzke and Botzet, 2007; Voigt and Marotzke, 2010; Voigt et al., 2011). Recent studies on the habitability of aqua-planets also indicate that the presence of multi-stability can

further complicate the interpretation of habitability of Earth-like planets (Linsenmeier et al., 2015; Boschi et al., 2013; Lucarini et al., 2013). With our study we demonstrate that under certain conditions Earth-like terra-planets exhibit two drastically different climate states and both these climate states can support habitable areas with long-lived surface liquid water.

It should be noted that the present paper is mainly descriptive in nature and is not meant to give a detailed
explanation of the mechanisms leading to the emergence of the two climate states and why transition happens between them. This is still under investigation. The paper is organised as follows: Sect. 2 describes our model, our terra-planet configuration and gives an overview on the simulations performed for this study. Sect. 3 discusses the two drastically different terra-planet climate states at perpetual equinox conditions. Sect. 4 describes the transition between the two climate states. In Sect. 5 hysteretic behaviour is discussed. Sect. 6 explores the role of the snow albedo feedback for
the emergence of the different climate states. Sect. 7 is about the terra-planet climate at present day obliquity. Sect. 8 draws some general conclusions from our study, in particular for the habitability of terra-planets.

## 2 Model and simulation setup

### 2.1 Model

We use the ICOsahedral Non-hydrostatic (ICON) General Circulation Model jointly developed by the MPI for
Meteorology and the German Weather Service (DWD) and run it in terra-planet configuration i.e. with a single globally extended continent. The model has a horizontal resolution of R2B04 equivalent to a resolution of an evenly distributed rectangular grid of about ~ 160 Km and 47 layers in the vertical. The atmosphere model uses a non-hydrostatic dynamical core on an icosahedral-triangular Arakawa C-grid (Zangl et al., 2015), and the model atmospheric physics is similar to ECHAM6 physics (Stevens et al., 2013). The radiative transfer calculations are based
on the Rapid Radiative Transfer Model (Mlawer et al., 1997; Iacono et al., 2008). Convection is parameterized by the mass flux scheme of Tiedtke (1989) with modifications to penetrative convection by Nordeng (1994). Cloud cover is calculated based on relative humidity (Lohmann and Roeckner, 1996). For a complete description of the model physics and parameterizations, the reader is referred to Stevens et al. (2013). The ICON version used in this study inherits the land physics from ECHAM5 (Roeckner et al., 2004) extended by a layered soil hydrology (Hagemann and Stacke,
30   2015).

### 2.2 Terra-planet configuration

The Terra-planet configuration is designed to be highly symmetric with no orography and glaciers. The rotation rate and the solar constant are the same as for the present day Earth. We consider two situations: perpetual equinox (zero obliquity) and present day Earth's obliquity (23.5°). Background concentration of $CH_4$, $N_2O$ and aerosols are fixed to
zero, while water vapor is prognostic and $CO_2$ concentration in the atmosphere is fixed to 348 ppmv. The ozone distribution is assumed to be zonally uniform and meridionally symmetric.

Land surface properties are assumed to be homogenous: The total soil depth in our study is about 10 meters (m) of which the layers below a depth of 1.2 m are forced to be filled permanently by at least 90% with water homogenously all over the globe. We refer to these bottom two layers of the soil in our study as 'subsurface reservoir'

which should not be confused with a geological reservoir operating at timescales much longer than the soil hydrological timescales. The root depth is fixed to 6 m such that the roots always have access to this subsurface reservoir. Leaf Area Index (LAI) is set to a value of 3. This value controls how much surface in a grid cell participates in transpiration, while the rest exhibits bare soil evaporation. LAI and root depth are not considered as a representation of vegetation but only as a technical means to parameterize the atmospheric access to water like other hydrological

parameters of the model, e.g. soil porosity or hydraulic conductivity. The albedo of snow ranges between 0.4 - 0.8 depending on the surface temperature. Surface roughness is fixed to 0.05 m.

By the introduction of the subsurface reservoir we implicitly equip our planet with a very efficient recycling mechanism shuffling water back from sink regions (P-E > 0) to source regions (P-E < 0). This can be understood as follows. In sink regions water either piles up as snow or is lost as runoff. But since neither snow height nor runoff

affect the climate in our simulations, the global amount of water relevant for the physical climate stays constant for a stationary state. Accordingly, considering only this 'effective' water, the amount of water added to the subsurface reservoir equals the water lost in the sink regions. For this reason, we can interpret the permanent refilling of the subsurface reservoir to mimic a very efficient recycling of water from sink to source regions.

Representing overland water recycling by a homogenously filled subsurface reservoir is indeed an

idealization. In fact, recycling may occur at different speeds and thereby is less or more effective than what we consider in our study. Based on the speed of recycling, the water level of this subsurface reservoir would vary from what is considered in our study. It should be noted that the choice of water level of the subsurface reservoir considered in our study (1.2 m) is not arbitrary, but a result of a sequence of simulations, where we explored the continuum between an aqua and a terra planet (see Appendix A).

Our terra-planet setup closely resembles that of recent 3-dimensional studies (Abe et al., 2011; Leconte et al., 2013) on Earth-like terra-planets except that those studies did not consider an overland water recycling pathway bringing back water from high to low latitudes that we mimic by the prescribed subsurface water reservoir.

**2.3 Simulations**

We study the effect of background surface albedo ($\alpha$) on the climate in a series of simulations with $\alpha$ varying from

0.07 to 0.24 at perpetual equinox (0°) (Z7 to Z24 simulations in Table 1) and at present day conditions (23.5°) (S7 to S24 simulations in Table 1). All these simulations start from the same initial atmospheric state with a homogenous temperature (290 K) and moisture content (25 kg m$^{-2}$). Due to the small thermal inertia of the land surface and the atmosphere, the simulations reach a steady state within 10 years. We continue the simulations for additional 30 years of which, the last 10 years are used in the analysis of the mean climate. To see the transition between different climate

states at the perpetual equinox condition more clearly, we perform an additional simulation (TRANS). For this, we first simulate the planet in the stable Hot and Dry (HD) state for 30 years for $\alpha$ = 0.14 and then increase $\alpha$ abruptly to $\alpha$ = 0.14 + 0.01 corresponding to the Cold and Wet (CW) state and continue the simulation for another 30 years. Then, to check for hysteresis, we continue the TRANS simulation by switching the albedo back to 0.14 and continue lowering $\alpha$ stepwise until zero. Additionally, to test the sensitivity of the climate states to model parameterizations,

we performed simulations with a different convection scheme at perpetual equinox conditions (T7 to T24 simulations).

Finally, to investigate the role of snow-albedo feedback on the terra-planet climate states, the terra-planet is simulated with dark snow (DS simulations in Table 1) (i.e. we assume snow albedo to be same as background surface albedo). The details of all terra-planet simulations are listed in Table 1.

## 3 Drastically different climate states

Figure 1 shows the time evolution of global mean surface temperature and precipitation for different terra-planet
simulations at zero obliquity (Z7 to Z24 simulations). We notice that the terra-planet exists in two drastically different climate states: a Hot and Dry (HD) state for $\alpha < 0.15$ and a Cold and Wet (CW) state for $\alpha \geq 0.15$. This is different to findings in previous studies (Abe et al., 2005, 2011; Leconte et al., 2013) on low obliquity terra-planets who found only the HD state. The mean climate in the two states is remarkably different (Fig. 2).

### 3.1 Surface Temperature
The annual mean surface temperature in the CW state is below freezing point almost everywhere, except in the low-latitudes where it is around 10°C. High cloud cover (Fig. 2e) and a high planetary albedo (Fig. 2f) lower the surface absorption of incoming radiation and result in very low temperatures in the CW state. On the other hand, in the HD state, the global mean surface temperature is about 35°C higher than in the CW state.

### 3.2 Precipitation
Besides temperature, the other striking difference between the two states is the location of precipitation bands on the planet. In the HD state, no precipitation occurs in the low-latitude region between 40°S to 40°N. Only latitudes higher than 40° receive some amount of rainfall (Fig. 2b) (compare Abe et al., 2005, 2011; Leconte et al., 2013). In contrast,
in the CW state, precipitation is mainly concentrated in the low latitudes with a banded structure similar to the present day equatorial Inter-Tropical Convergence Zone (ITCZ). The reason for the absence of CW state in previous studies is the lack of an effective mechanism that can recycle the water trapped at the high latitudes in the form of snow and ice back to the low latitudes.

### 3.3 Feedbacks that keep HD State dry and CW State wet
Figures 2c and d show the distribution of water on the planet. In the HD state, the very high temperatures in the low latitudes raise the water vapour saturation limit and the moisture-holding capacity of the atmosphere allowing the planet to store a substantial amount of its water in the atmosphere (Fig. 2c). In such an atmosphere, rain occurs in the form of virga rain i.e. almost all of this rain evaporates on its way to the surface. Therefore, in such a case, rain does
not contribute to the moistening of the soils in the low-latitudes. This, along with huge amounts of net radiation at the surface in the HD state keeps the uppermost soil layers in the low latitudes very dry. Dry uppermost soil layers imply small evaporation leading to no precipitation which in turn leads to even less evaporation. This self reinforcing mechanism in the HD state always maintains very dry upper soil layers in the low latitudes (Fig. 2d). On the whole, supressed precipitation at low latitudes along with a nevertheless permanent export of moisture away from the low
latitudes result in water to be mainly present at high latitudes in the HD state. On the other hand, in the CW state, the

lower annual mean temperatures in the low latitudes facilitate condensation and precipitation and minimise the moisture content of the atmosphere (Fig. 2c). Evaporation of falling rain indeed also occurs in the CW state, but is not strong enough to prevent the rain from reaching the surface. Thus, continuous precipitation at low latitudes in the CW state, keeps the upper soil layers in the low latitudes always wet (Fig. 2d) thus providing by evaporation sufficient water for a stable precipitation regime in the low latitudes.

**3.4 Feedbacks that keep HD State hot and CW State cold**

The vertical distribution of cloud cover and cloud water content in the low latitudes for the two terra-planet states is displayed in Figure 3. For the HD state, the cloud cover in the low latitudes is exclusively composed of high level clouds (Fig. 3). The reason is, the higher water vapour saturation limit of the atmosphere in the HD state raises the

height at which condensation and cloud formation occurs. High clouds with very low liquid water content are more transparent to shortwave radiation, at the same time they reduce outgoing longwave radiation and thereby warm the planet. Moreover, hotter temperatures in the HD state lead to moister atmosphere (Fig. 2c) and in turn stronger greenhouse warming. Overall, high clouds in the low latitudes along with higher water vapour greenhouse warming keep the planet always hot and stabilize the HD state. Instead, in the CW state, the cloud cover in the low latitudes is

mainly comprised of low level clouds (Fig. 3) due to lower water vapour saturation limit of the atmosphere. Low clouds with high liquid water content cool the planet as they increase the planetary albedo. Also, lower temperatures in the CW state lead to a drier atmosphere (Fig 2c) and a weaker water vapour greenhouse warming. On the whole, low clouds together with weaker greenhouse warming always keep the planet cool and stabilize the CW state.

**3.5 Mean Circulation and energy transport**

The mean circulation pattern for the two climate states is shown in Fig. 4a, b. We notice that in both the states, the circulation pattern resembles the present-day hemispheric three cell structure. But when comparing the two states, the width and intensity of the circulation are very different: In the HD state, the Hadley cell is more vigorous and gets slightly wider with height (when measured around 500hPa) compared to that in the CW state. The neutrally stable

atmospheric conditions found in the HD state require a larger mass flux to transport away heat and to stabilise the equatorial temperatures and hence support a more vigorous circulation (Held and Hou 1980; Caballero et al. 2008; Mitchell et al., 2008). Additionally, we notice that the circulation centre of the Hadley cell is very different in the two states (depicted by black arrows in Fig. 4 a,b). The HD state has its circulation centre much closer to the surface at about 750 hPa, while in the CW state the centre is much higher at 500 hPa.

Observations and modelling studies in the literature also report intense low centred circulations for present day Earth's climate in the tropical eastern pacific during the northern hemisphere summer time (Zhang et al. 2004, 2009; Nolan et al., 2007), for a Snowball Earth state (Pierrehumbert, 2005) and for Titan (Mitchell et al., 2006, 2009). These circulations have the tendency to export larger amounts of moisture out of the low latitudes compared to high centred circulations. We also notice this increased transport of moisture away from the low latitudes with the more vigorous

low centred Hadley cell in the HD state compared to that in the CW state (Fig. 4c). The reason is, in the CW state the existence of deep convection allows moisture to reach much higher elevations before getting exported to high latitudes

(Nolan et al., 2007). At higher elevations, more moisture can condense and be lost as precipitation due to low temperatures and thus much less water remains for being exported away. Instead, in the HD state the lack of precipitation allows more moisture to be exported.

Figure 5 shows the northward transport of energy (transport due to latent and sensible energy) by atmospheric circulation in the two climate states. For the CW state, latent energy transport is equatorward in the low latitudes and 10  poleward in the mid latitudes. In contrast, in the HD state latent energy is exported poleward at all latitudes (latent energy curves in Fig. 5). The reason for the opposite sign in the low latitudes is that in the CW state (as in the case of present day Earth) intense precipitation dries out the atmosphere and thus keeps most of the water within the low latitudes limiting its poleward export (Fig. 4c) – hence at these latitudes the net flow is always equatorward. Instead, in the HD state, by the absence of precipitation, moisture is retained in the air so that it is exported poleward.

The sensible energy transport in the HD state is considerably larger as compared to that in the CW state (Fig. 5) despite of a lower equator to pole temperature gradient. This implies that the atmospheric circulation is more efficient in transporting the sensible energy to the high latitudes. Nevertheless, the net northward transport of energy is dominated by the sensible energy transport. Further, the larger northward transport of energy in the HD state results in a smaller equator to pole temperature gradient compared to the CW state.

**4 Transition to the CW state**

Starting from the HD state ($\alpha$ = 0.14),  we abruptly increase the albedo to $\alpha$ = 0.14 + 0.01 (simulation TRANS) and thereby initiate a shift to the CW state. The full transition from the HD state to the CW state takes about 4 years. The changes in annual mean surface temperature, precipitation, snow cover and mean circulation during the course of transition are shown in Fig. 6. One can distinguish three transitional stages:

**Stage 1:** After the abrupt increase in $\alpha$, small precipitation clusters appear in the low-latitude region and the terra-planet starts to cool down due to an increase in cloud cover and planetary albedo. Snow cover at high latitudes starts to increase slowly. The mean meridional circulation structure changes – small circulation cells appear very close to the equator. This happens for a period of one year.

**Stage 2:** The precipitation clusters aggregate into a band of deep convection around the equator accompanied by a 30  sharp increase in precipitation. At this point, we still see precipitation bands even at high latitudes around 50°. However, the precipitation intensity at high latitudes is smaller (around 6 mm day$^{-1}$ lower) compared to the precipitation at the equator. Surface temperature drastically decreases and the circulation structure is now associated with two cells, a shallow cell close to the equator and a deep cell slightly away from the equator.

**Stage 3:** Precipitation bands at the high latitudes start moving equatorward and precipitation intensity decreases 35  compared to the previous stage. Surface temperature decreases further. Snow cover further increases and starts moving towards the equator. The circulation is now less intense with circulation centre around 500hpa.

Finally, the CW state is reached. Precipitation only occurs at low latitudes. Surface temperature is below freezing almost everywhere on the planet except at the low latitudes. Snow cover reaches down to the 40° latitude.

 **5 Hysteresis**

To further study the bifurcation structure, we investigated the hysteretic behaviour. Figure 7 shows the global mean surface temperature plotted as a function of $\alpha$. The spontaneous transition from the HD to the CW state and the associated abrupt cooling is seen as path 1. Starting from the CW state and lowering back $\alpha$ stepwise below the threshold value until zero does not lead back to the HD state. The reason is that the high cloud cover present in the CW state hides the surface from solar irradiation. Therefore, a reduction of $\alpha$ has only a minor effect on the top of the atmosphere radiation balance so that thereby it is impossible to heat the planet sufficiently strongly to switch back to the HD state. This is true even when repeating these simulations taking snow albedo equal to background soil albedo (experiments DS; no figure shown). Thus, the planet remains in the CW state indicating that under the chosen conditions for the terra-planet setup the hysteresis is not closed (Fig. 7).

**6 Does snow albedo feedback play a role for the emergence of the two climate states?**

Changes in snow cover can lead to multiple climate states in the Earth system with drastically different global mean surface temperatures like the present day Earth and the Snowball Earth (Budyko 1969; Sellers 1969). In this case, the large temperature difference between the two states is caused by the positive snow-albedo feedback. In our study, we also notice such huge differences in global mean surface temperatures between the two terra-planet climate states (Sect. 3). In order to test whether the snow albedo feedback is responsible for the huge temperature difference in our study, we performed additional simulations where we changed the snow albedo to be equal to the background albedo of soil (simulations DS in Table 1). With the darker snow, we still find the two climate states (Fig. 8) and the spontaneous transition between them. The existence of the bifurcation even in the simulations with dark snow implies that the snow-albedo feedback is not the cause for the existence of the two states. However, the snow albedo feedback does enhance the drastic temperature change in the bright snow simulations by around 12°C compared to the dark snow simulations. Preliminary analysis indicates that a combination of cloud and hydrological feedbacks leads to the bi-stability (refer Sect 3.3 and 3.4).

**7 Terra-planet with seasonality**

So far, we considered a planet without seasonality because obliquity was set to zero. Next, we investigate the climate on a terra-planet changing with a seasonal orbit taking present day Earth's obliquity of 23.5° (S simulations in Table 1). We find that with seasonality the HD state is absent and the terra-planet always stays in the CW state (Fig. 9). This is contrary to previous studies on terra-planets which show that terra-planets always exist in a warm state for obliquities lower than 30° (Abe et al., 2005, 2011). We hypothesize that for Earth-like obliquity, the bi-stability is lost due to the seasonal migration of the rain bands towards the low latitudes. The seasonal migration of rain bands facilitates seasonal rain in the dry low-latitude region. Since soil moisture has a memory lasting for several weeks to months this causes the top soil layers in the low latitudes to remain wet even during the dry season. Thus there is always soil moisture in the originally dry low-latitude region to allow for continuous evaporation and precipitation. And this probably destroys the HD state so that the planet is always self stabilized in the CW state. A more detailed

study on the reasons behind the loss of bi-stability for non-zero obliquities is ongoing but beyond the scope of the present paper.

## 8 Discussion and Conclusions

So far terra-planets have been investigated for a wide range of planetary properties like mass, rotation rate, atmospheric composition and orbit (Abe et al., 2005, 2011; Leconte et al., 2013). Here, we focus on climate simulations of a terra-
planet with low obliquity (< 30°), flat orography, and otherwise Earth like conditions. An important difference to previous studies concerns the treatment of atmospheric access to water. Past studies on possible climates of terra-planets prescribed a limited water inventory, which for low obliquities leads to trapping of the water at high latitudes stabilizing the planet in a state with no precipitation at low latitudes, similar to what we call in our study 'Hot and Dry' state. By contrast, in the present study we assume an unlimited subsurface water reservoir, which is still different from
an aqua planet configuration (having also an unlimited water reservoir), because resistances between soil and atmosphere restrict the atmospheric access to soil water in addition to the restriction from aerodynamic resistance. For such an Earth-like terra-planet with restricted water access we find two drastically different climate states, a Hot and Dry (HD) state characterized by a hot climate with precipitation confined to high latitudes and a Cold and Wet (CW) state which is closer to present-day Earth's climate with precipitation mainly occurring at low latitudes and an intense
cycling of water there. Compared to the other studies mentioned above, we find this additional CW state only, because by prescribing the subsurface water reservoir we implicitly assume for our terra-planet a mechanism restoring water from the high latitudes to low latitudes, refilling the subsurface water reservoir sufficiently effective to maintain the very active low latitude water cycle in this state. The difference in global mean temperature between these two climate states is 35°C (with the same boundary conditions), which is in the same order of magnitude as the temperature
difference between present day and the Snowball Earth climate (Pierrehumbert et al., 2011; Micheels and Montenari, 2008; Fairchild and Kennedy, 2007; Hoffman and Schrag, 2002).

Similar to the abrupt transition to a Snowball Earth state, for perpetual equinox conditions our terra-planet simulations also show an abrupt transition, namely from the HD state to the CW state. These two states exist for low background surface albedo $\alpha$, while for high $\alpha$ only the CW state is possible. The abrupt transition occurs when in the
HD state $\alpha$ is increased beyond a particular threshold value. While the transition to the Snowball Earth is driven by the snow albedo feedback, preliminary analysis indicates that in our study the transition is triggered by a re-organisation of the hydrological cycle and amplified by cloud feedbacks. Moreover, we notice that in our setup the terra-planet exhibits an open hysteresis: even with background surface albedo reduced to zero it does not return back to the HD state. Additional terra-planet simulations with an obliquity as the real Earth all result in a CW state, hinting
at another bifurcation from the HD to the CW state when the obliquity is increased to non-zero values.

Concerning the global water cycle, the HD and CW states share one important similarity namely a strong atmospheric moisture flux from low to high latitudes. The planetary boundary layer at low latitudes is extremely dry in the HD state with relative humidity of about 15%. Clearly, trade winds can maintain such a dry boundary layer only if the water supply at the surface is sufficiently limited. However, without any evapotranspiration at the surface the
Hadley Cell would dry out completely, loosing the greenhouse effect that sustains the high temperatures of the HD

state. Furthermore, in the CW state, to keep the rain along the equator one also needs a considerable water supply in the low latitudes. In summary, both climate states are associated with a strong atmospheric transport of water from low to high latitudes, which has to be balanced by evapotranspiration at the low latitudes. Therefore, allowing for both states to be potentially realized under the same boundary conditions, on a real planet, mechanisms must exist that can continuously restore water back to low latitudes. For present day Earth this happens via the oceans. For our terra-

planet water is stored in frozen form at high latitudes like in the past glacial states of our Earth. The resupply of water may happen via processes like melting of glaciers, transport of ice by gravity flows and melting at the bottom of large ice caps due to high pressure and geothermal heat flux. This provides liquid water, which may be brought back to the low latitudes by rivers (Abe et al., 2010; Leconte et al., 2013). Note that the huge differences in climate between the two states is primarily a consequence of the completely different functioning of the global hydrological cycle, so that

the assumed recycling mechanism can be considered as an additional degree of freedom to the internal dynamics extending the range of possible terra-planet climates.

Our findings may have some relevance for estimates of the habitable zone for such Earth-like terra-planets. In the HD state liquid water is confined to the mid-latitudes (40°-50°) in both hemispheres, whereas in the CW state, there are sufficient precipitation and high enough temperatures for permanent liquid water at low latitudes (40°S -

40°N). Thus in both climate states life can potentially persist. At the outer edge of the habitable zone, the assumed resupply of water from high to low latitudes stabilizes the greenhouse effect, keeps the planet in the HD state and may prevent a situation with all water accumulated at the high latitudes in the frozen form. At the inner edge of the habitable zone, by this resupply the planet can maintain precipitation and high cloud cover at the equator in the CW state. Thereby, the planetary albedo is increased which cools the planet and may prevent the runaway greenhouse state with

all the water well mixed in the atmosphere in the gas phase. On the whole, our study thus suggests that the presence of a mechanism which recycles water from the high latitudes back to the low latitudes results, as described, in the two drastically different climate states and may extend the habitable zone of Earth-like terra-planets at low obliquities.

**Appendix A: Sequence of simulations that led to the terra-planet bi-stability**

The bi-stability found in this study should not be mistaken as a result of a random experiment or a bug in the land

surface model. The bi-stability was found by systematically modifying our global climate model starting from a "swamp simulation" until a "terra planet configuration". The respective simulations showed climate states that were plausible from the configuration changes. In the "Swamp configuration" our model is able to successfully reproduce the climate of an "Aqua-planet". Starting from this setup, we sequentially changed different parameters like surface heat capacity, surface roughness, background soil albedo, snow albedo, and the level of the global subsurface water

reservoir as well as the land surface schemes (soil hydrology) in our land surface model (Table A1). The bi-stability emerged when the water level of the subsurface reservoir was lowered to 1.2 m. In Table A1 the sequence of simulations is listed and in Fig A1 we show the resulting time evolution of temperature and the zonal structure of precipitation.

## Appendix B: Sensitivity of the climate states to model convection scheme

To confirm that the two climate states of the terra-planet are not an artifact of convective parameterizations, we performed an additional simulation with a different convection scheme. By default, the model uses the Nordeng convection scheme (Nordeng 1994). We have replaced the default settings and simulated terra-planet simulations with the Tiedtke convection scheme (Tiedtke 1989) and we find the two drastically different climate states irrespective of the convection scheme (Fig. B1).

**Competing interests.** The authors declare that they have no conflict of interest.

**Acknowledgements.** We thank the reviewers for their valuable inputs to the manuscript. We also thank Reiner Schnur, MPI-M for the technical support and Jürgen Bader, MPI-M for his valuable suggestions. This work was supported by the International Max Planck Research School on Earth System Modelling (IMPRS-ESM) and the Max Planck Society (MPG). The computational resources were provided by the Deutsches Klima Rechenzentrum (DKRZ). The article processing charges for this open-access publication were covered by the Max Planck Society.

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

| Simulations | Background soil albedo | Obliquity | Convection scheme | Snow albedo |
|---|---|---|---|---|
| **Z7 – Z24** | 0.07, 0.12, 0.14, 0.15, 0.17, 0.24 | 0 | Nordeng | Dynamic |
| **S7 – S24** | 0.07, 0.12, 0.14, 0.15, 0.17, 0.24 | 23.5 | Nordeng | Dynamic |
| **TRANS** | 0.14 – 0.15 – 0.14 – 0.12 – 0.07- 0.00 | 0 | Nordeng | Dynamic |
| **T7 – T24** | 0.07, 0.12, 0.14, 0.15, 0.17, 0.24 | 0 | Tiedtke | Dynamic |
| **DS** | 0.07, 0.12, 0.14, 0.15, 0.17, 0.24 | 0 | Nordeng | Constant at background soil albedo |

5  **Table 1. Summary of simulations performed in this study.**

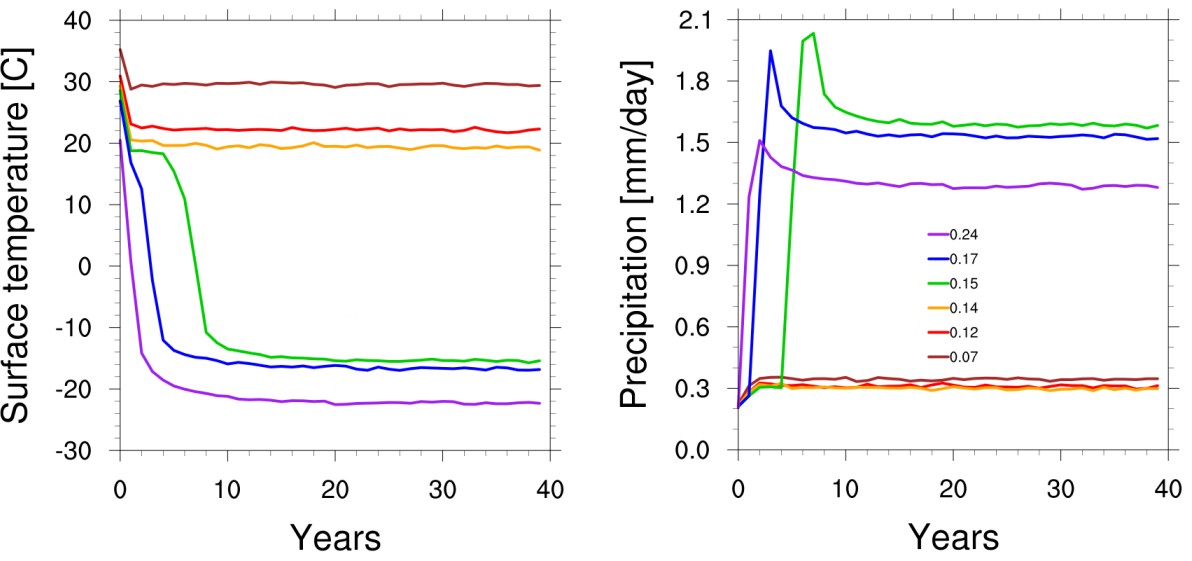

**Figure 1: Time series of global mean surface temperature (°C) and precipitation (mm day⁻¹) for different background soil albedo values in the Z7- Z24 simulations.**

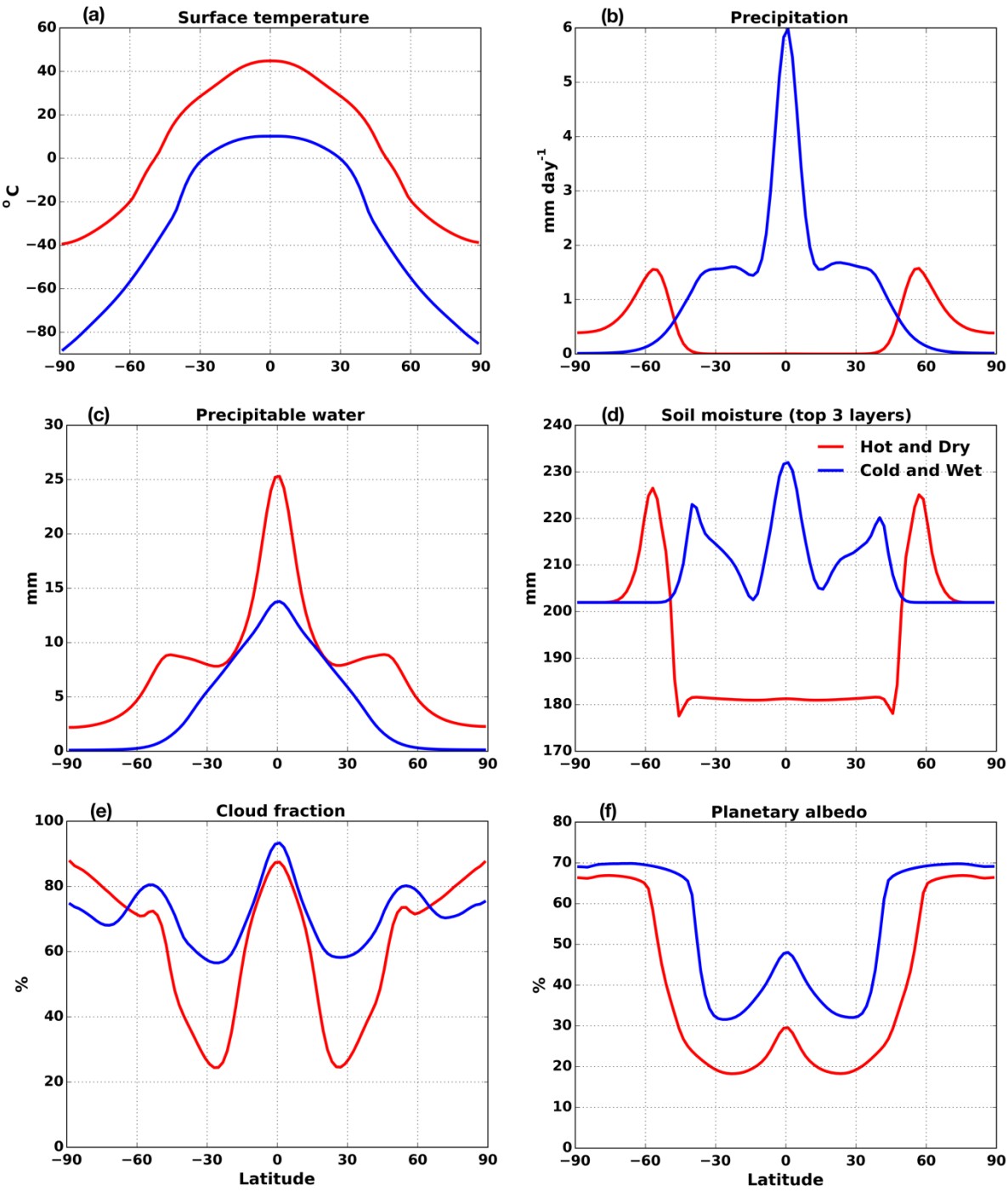

**Figure 2: Annual mean meridional profiles of (a) surface temperature, (b) precipitation, (c) precipitable water, (d) soil moisture (averaged over the top three layers), (e) cloud fraction and (f) planetary albedo for the two terra-planet states: HD ($\alpha$ = 0.14) and CW ($\alpha$ = 0.15) in the simulations Z14 and Z15 averaged over a period of ten years.**

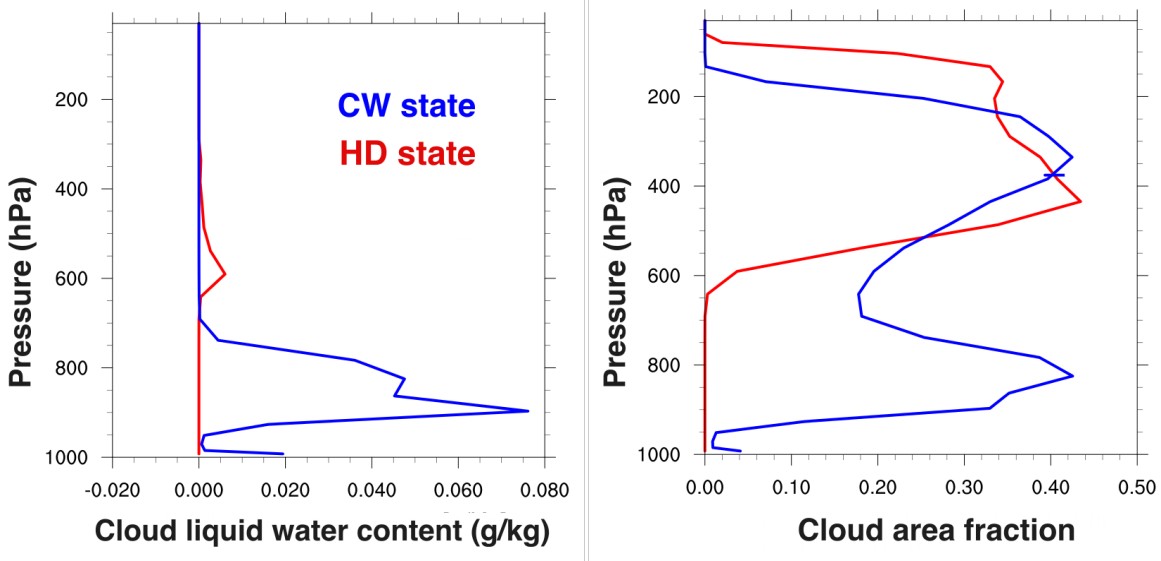

**Figure 3: Vertical profiles of cloud liquid water content (g kg$^{-1}$) and cloud area fraction averaged over the low latitude region (30°S - 30°N) for the two terra-planet states in the simulations Z14 and Z15.**

25

30

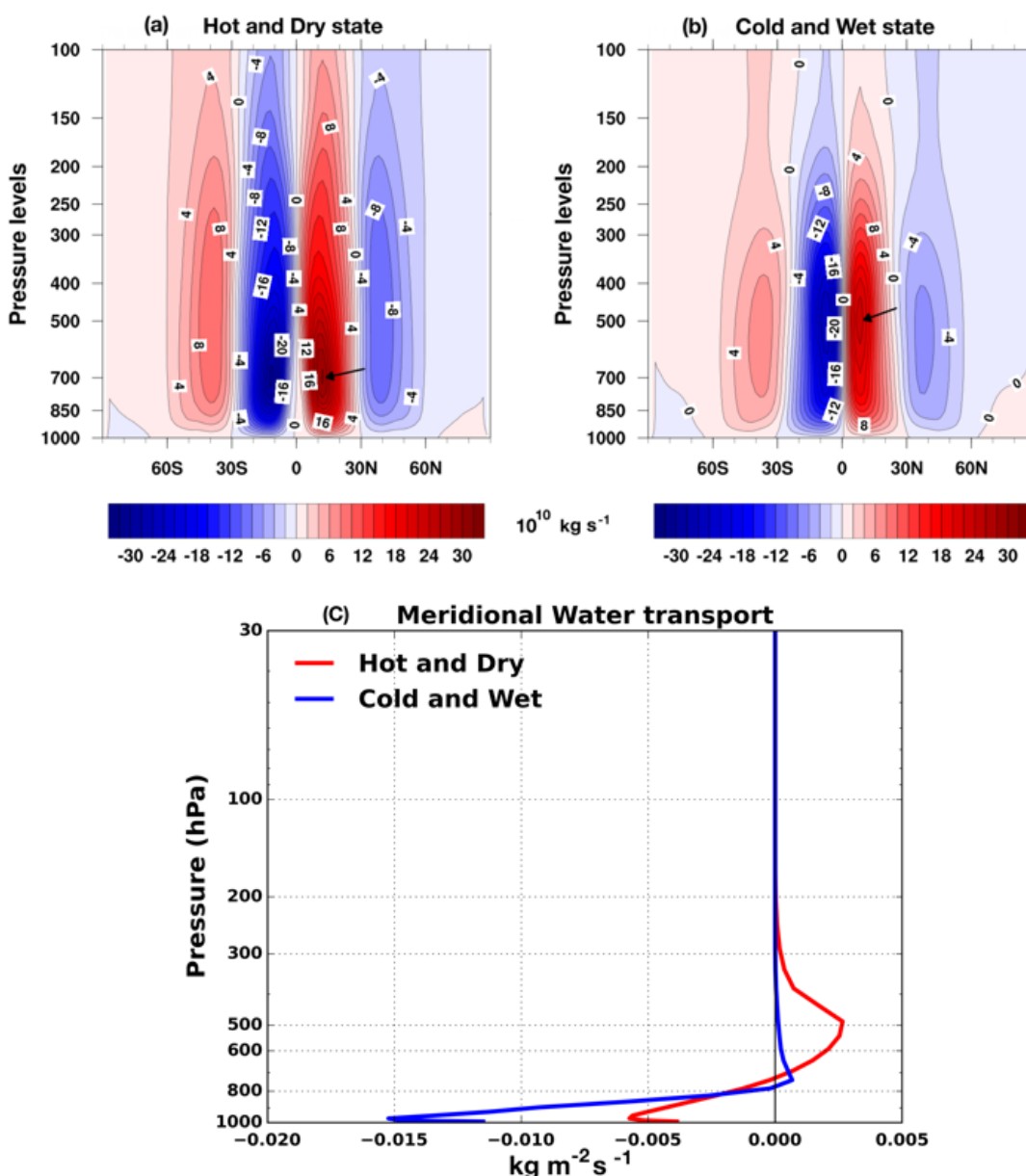

**Figure 4: Annual mean behavior of (a) and (b) meridional stream function in $10^{10}$ kg s$^{-1}$, (c) vertical profiles of meridional transport of water at 10°N for the two terra-planet states in the simulations Z14 and Z15 averaged over a period of ten years. The black arrows in (a) and (b) denote the northern Hadley circulation centre.**

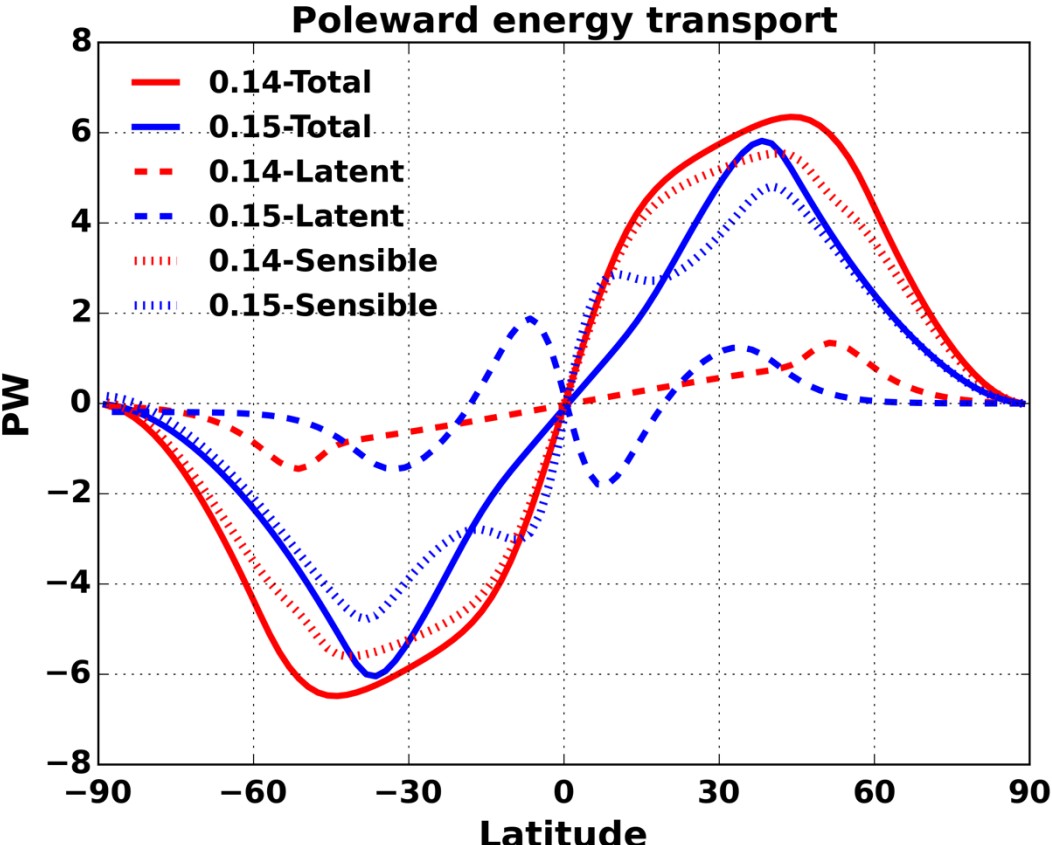

**Figure 5: Annual mean behavior of northward transport of total, latent and sensible energy in peta watts (PW) for the two terra-planet states in the simulations Z14 and Z15 averaged over a period of ten years.**

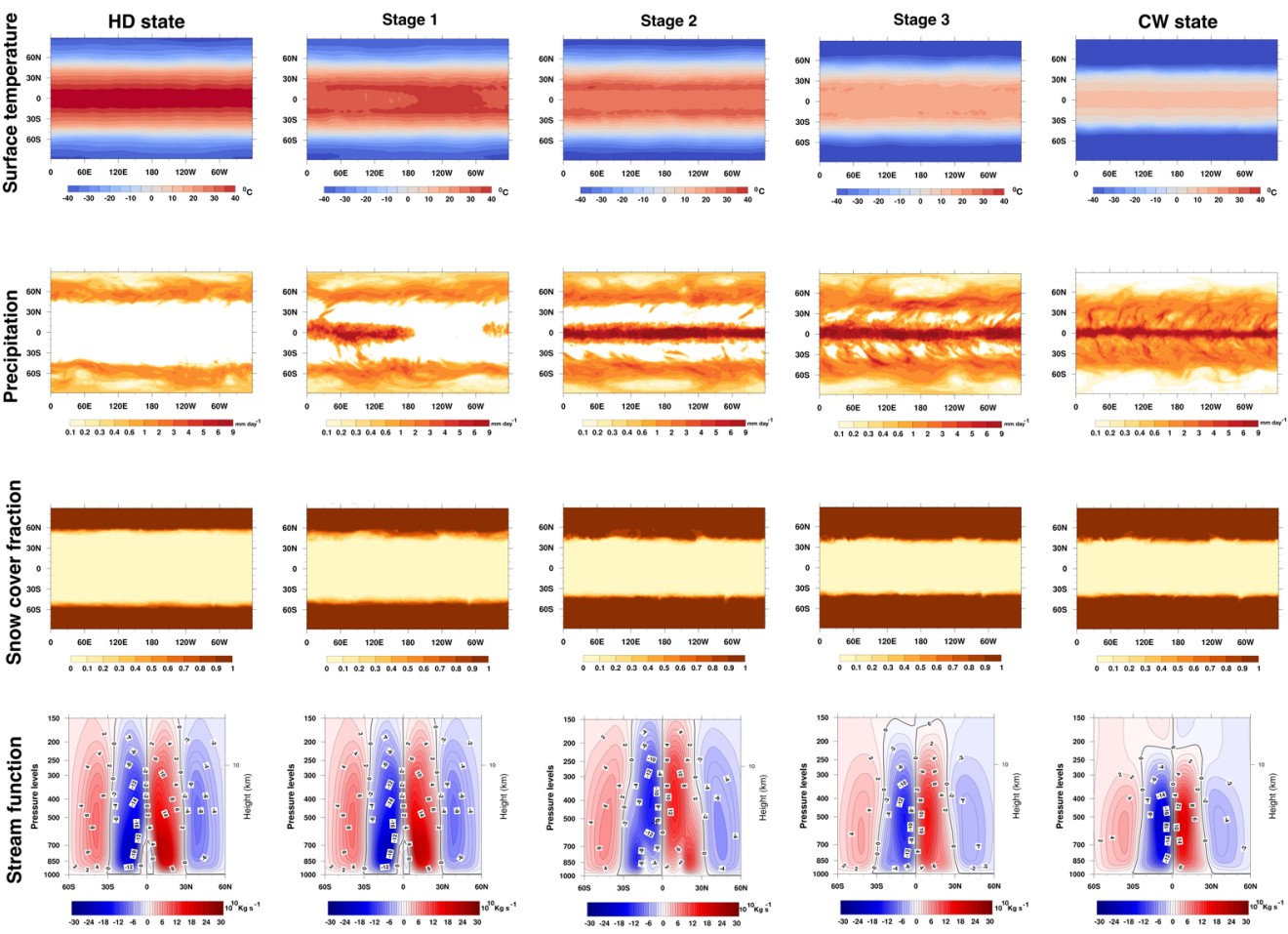

**Figure 6: Annual mean surface temperature (first row), precipitation (second row), snow cover fraction (third row) and meridional stream function (fourth row) for different stages of the transition (TRANS simulation) from the HD state to the CW state.**

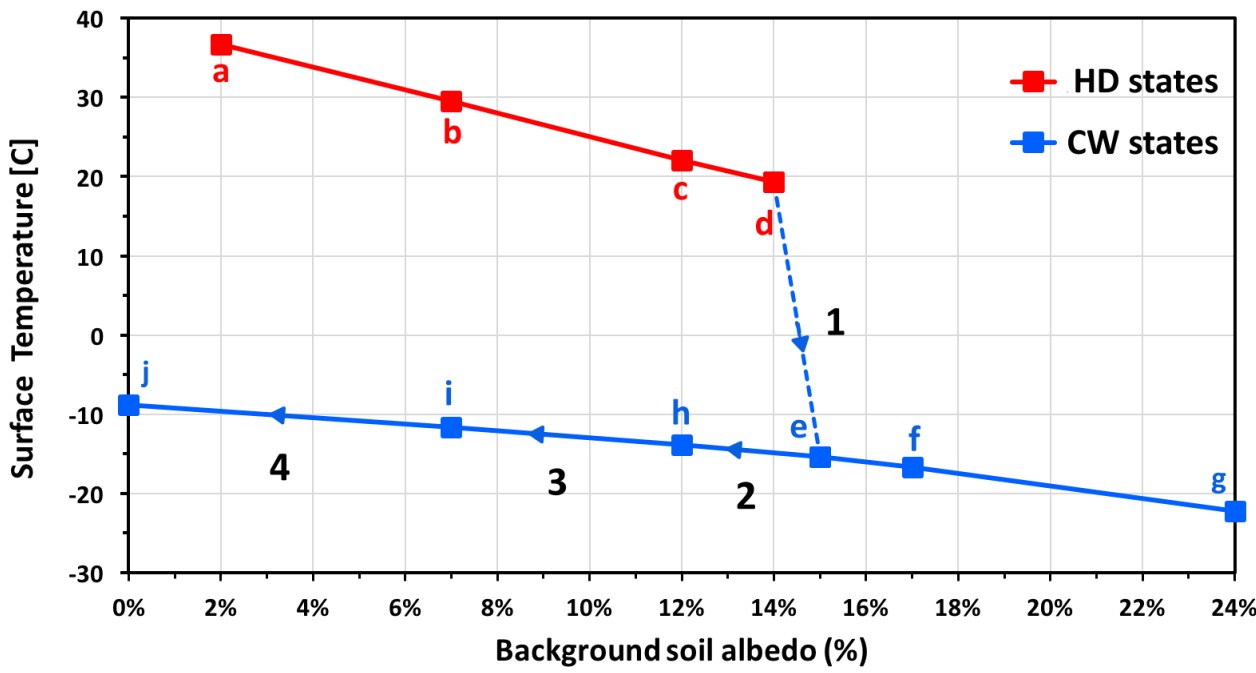

**Figure 7: Global mean surface temperature as a function of background soil albedo ($\alpha$). Simulated HD states are denoted by points a-d and simulated CW states by e-j. Path 1 denotes the spontaneous transition from HD to CW when increasing $\alpha$ and paths 2, 3 and 4 denote the reverse state development upon stepwise lowering of background albedo starting from the threshold value until zero. Obviously, the hysteresis doesn't close for the terra-planet at zero obliquity considered here.**

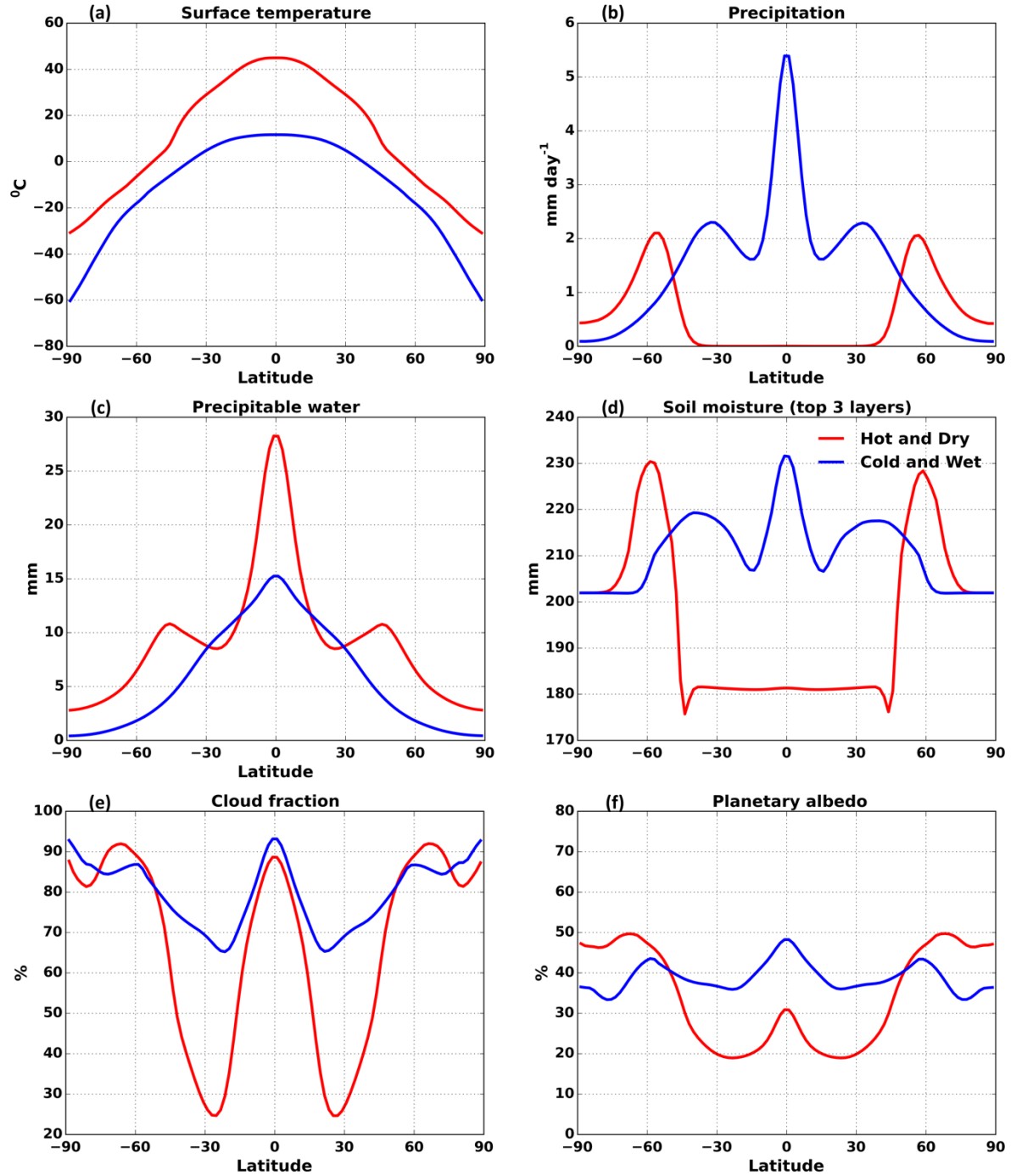

**Figure 8: Annual mean meridional profiles as in Fig. 2 but with dark snow (snow albedo is same as background soil albedo – DS simulation).**

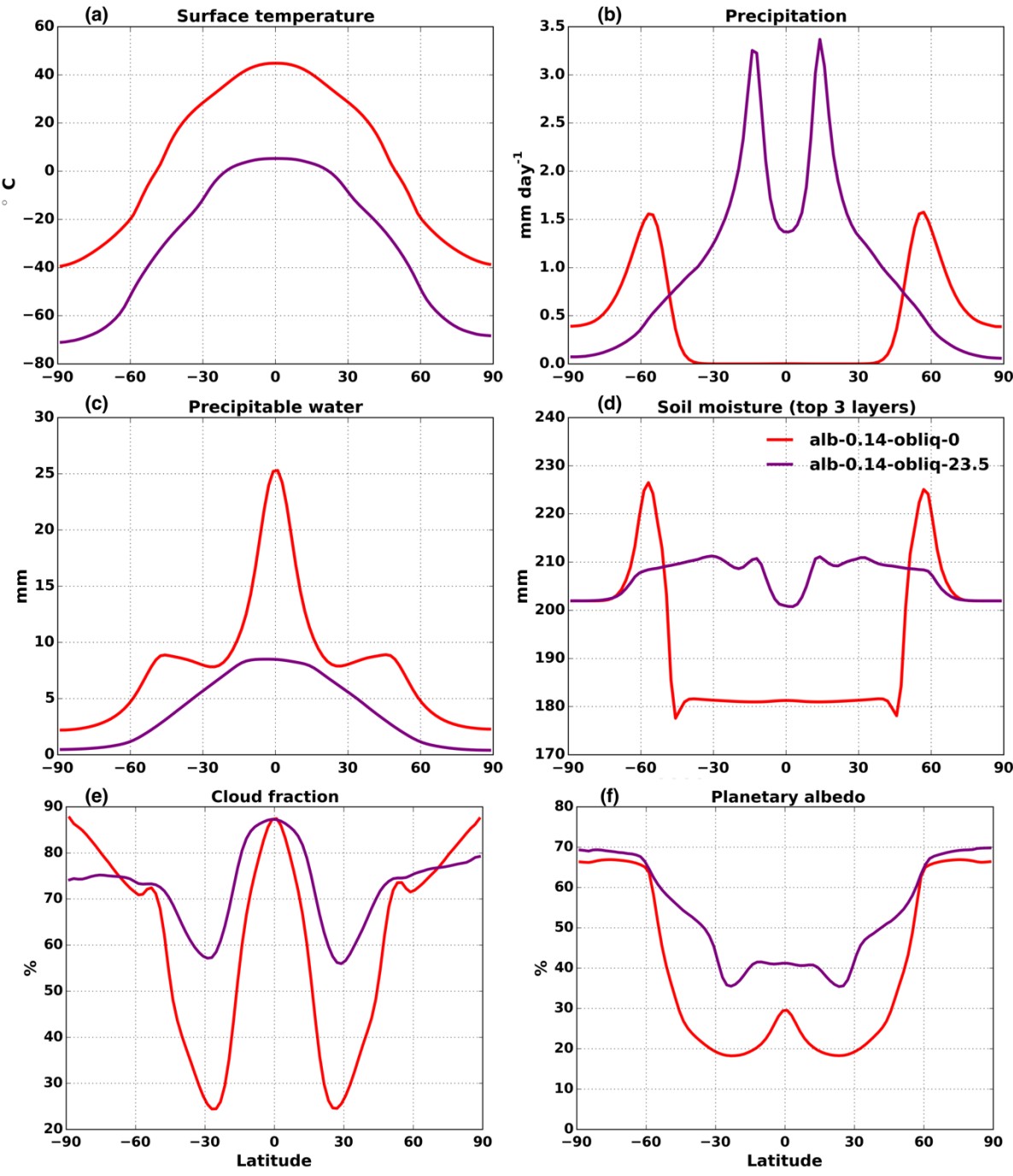

**Figure 9: Annual mean meridional profiles as in Fig. 2. The red lines represent the HD state as in Fig. 2 (Z14 simulation in Table 1 with an obliquity of 0°). The purple lines correspond to the simulation with same initial conditions as in the HD state in Fig. 2 but with an obliquity of 23.5° (S14 simulation in Table 1).**

| Simulations | Background soil albedo | Surface roughness | Heat capacity | Snow albedo | Water reservoir depth |
|---|---|---|---|---|---|
| **Aqua-planet** | 0.07 | Ocean | Ocean | 0.07 | 50 m slab ocean, no heat transport |
| **Swamp1** | 0.07 | Ocean | Ocean | 0.07 | Constant ground water table at a depth of 0.3 m |
| **Swamp2** | 0.07 | Land | Land | 0.07 | Constant ground water table at a depth of 0.3 m |
| **HD** | 0.07-0.14 | Land | Land | Dynamic (0.4 -0.8) | Constant ground water table at a depth of 1.2 m |
| **CW** | 0.15-0.24 | Land | Land | Dynamic (0.4 -0.8) | Constant ground water table at a depth of 1.2 m |

**Table A1. Summary of simulations performed to illustrate the procedure followed by which we found bi-stability on our terra-planet.**

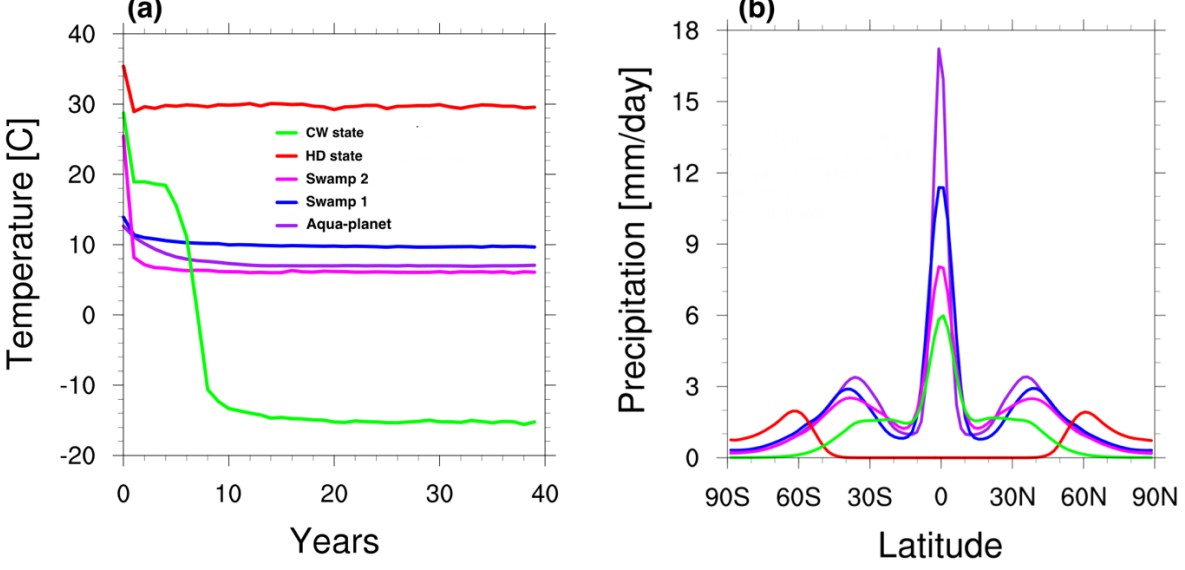

15 **Figure A1: (a) Time series of global mean surface temperature (°C) and (b) annual mean meridional profile of precipitation (mm day[-1]) for different simulations performed to illustrate the procedure by which we found bi-stability on our terra-planet.**

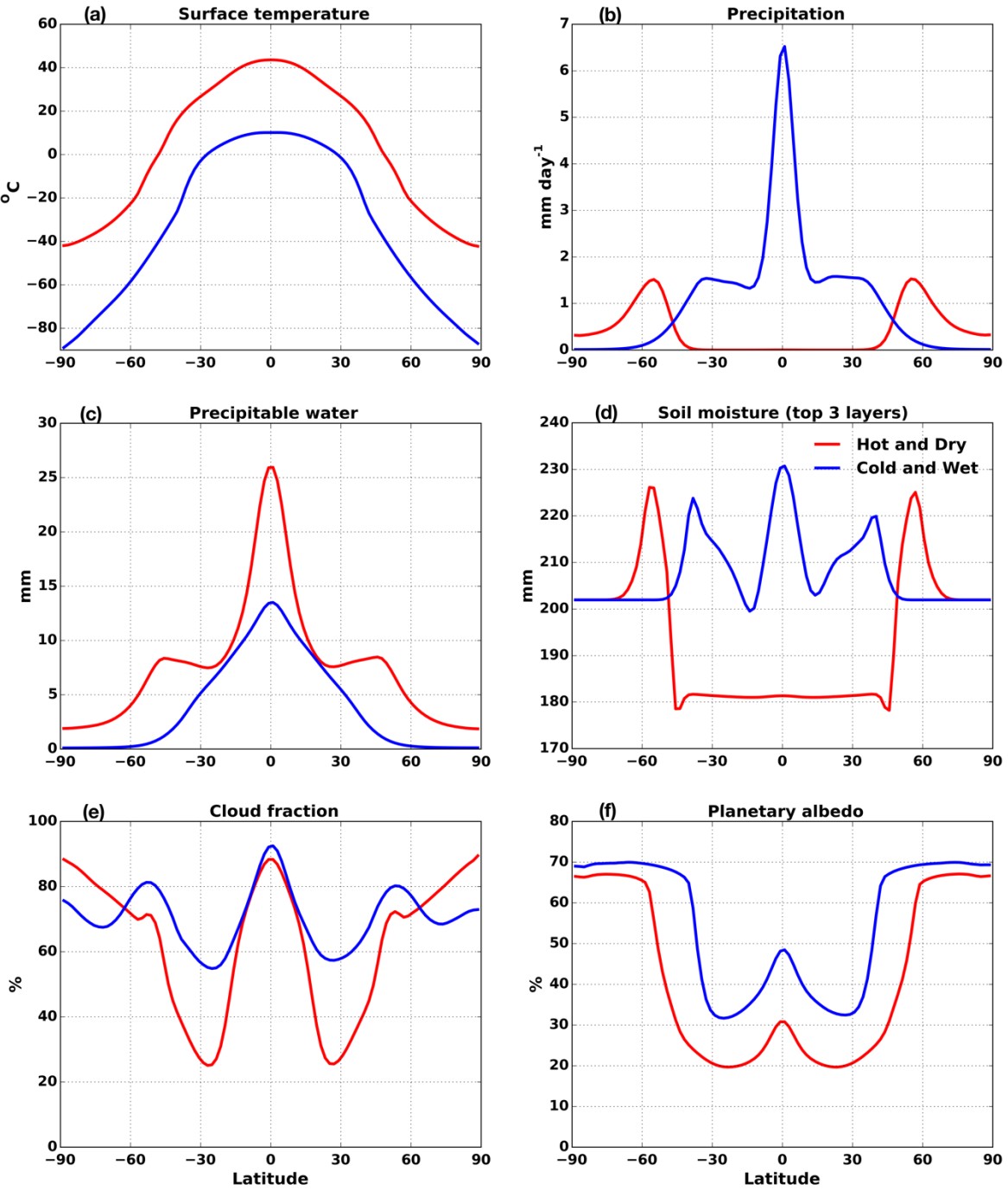

**Figure B1: Annual mean meridional profiles as in Fig. 2 but with Tiedtke convection scheme plotted using T simulations in Table 1.**