# Peer review of "Two drastically different climate states on an Earth-like terra-planet"

_Earth System Dynamics, 2017_

## Referee Comment (RC1) · D.S. Abbot (Referee) · 11 Oct 2017

**Paper:** Two drastically different climate states on an Earth-like terra-planet
**Authors:** Kalidindi et al.
**Journal:** ESD
**Reviewer:** Dorian S. Abbot
**Date:** October 10, 2017

**Overview:** The authors run a sophisticated global climate model (GCM) with a surface that is covered by a sophisticated land model. They assume a globally uniform ground water table, representing efficient surface water transport. Interestingly, they find bistability between two climate states, a hot and dry (HD) state and a cold and wet (CW) state. I have never seen anything like this before and it is definitely worth publishing. I think the authors should do a bit more work to understand why the model produces this behavior, then the paper will be ready to be published. I have some comments that hopefully will help with this.

**Comments:**

**1. Mechanism:** In the final paragraph of the introduction the authors restrict the scope of the paper so that it doesn't include investigating the mechanism of bistability. I think at least some investigation of the mechanism is necessary so that this paper stands as an independent work that doesn't rely on future work and to convince the reader that the observed behavior isn't simply due to some bug in the land surface scheme.

The way I would approach this is to first run the model in "swamp" configuration: with a global mixed layer ocean of 1 m depth and zero ocean heat transport. Presumably you will reproduce a roughly Earth-like climate. Then I would turn on the land surface model, but choose schemes and parameters to match the swamp configuration as closely as possible. Then slowly turn more schemes on or change parameters until you get bistability. This should allow you to identify the specific physical parameterization that allows for bistability. It must be related to evapotranspiration, since that's the key difference between a swamp surface and a real land surface.

**2. Surface energy balance:** Surface energy balance is important for global-mean evaporation, and therefore precipitation. All else equal, more absorbed shortwave at the surface should mean more evaporation and more precipitation. That's why it seems odd that the planetary albedo should be lower in the HD state, but the global-mean precipitation should also be so much lower. One possibility is that much more of the surface heat is lost through sensible rather than latent heat in the HD state. Another is that the planetary shortwave absorption is not a good proxy for surface shortwave absorption because of differential atmospheric absorption of radiation in the two cases. I think it's worth calculating the terms in the surface energy balance (both zonal mean and global mean) and using this to explain why the HD state seems to be able to absorb so much more shortwave yet have such a low evaporation. If you can explain this, it might also help your investigation into the mechanism for bistability. Also, I wonder if this could be connected to why bistability is lost for Earth-like obliquity. It would be really great if you could explain why increasing obliquity disrupts the bistability, and maybe analyzing the surface energy balance would help.

**3. Vertical Temperature Structure:** One thing I was wondering about is the vertical temperature structure in the two states, since convection is probably important for the bistability. I think it

would be worth plotting and thinking about this. A related point is that near the outer edge of the habitable zone we expect very high $CO_2$ levels, probably at least 1 bar. A large radiative cooling in the atmosphere could strongly affect the vertical temperature structure. It would be interesting to do some test runs at very high $CO_2$ and see if the HD state can persist under these conditions, since the authors connect the HD state with the outer edge of the habitable zone.

---

## Referee Comment (RC2) · Anonymous Referee #2 · 16 Nov 2017

**General comments**

In this paper, the authors modelled the climate of a terra planet (or land planet), a planet with a global land surface, without ocean. The particularity of the present study is the presence of an unlimited underground water layer providing water at all latitude but behaving differently as an ocean. The authors perform various simulations, changing obliquity, snow albedo and a test with a different convection scheme.

Depending on the value of the surface albedo, two different stable climate states apear. A hot and dry state and a cold and wet state. This latter is a novelty, due to the addition of the underground water reservoir and is an interesting result.

However, as the main novelty is the presence of an underground reservoir, more discussion would be useful. For instance the authors tested only a fixed depth of 1.2m to mimic a "recycling" of water from higher to lower latitudes. This setup is somewhat artificial and a justification of the current assumptions as well as the potential effect of varyng them would be welcomed.

**Specific comments**

lines 29-30 of page 4. The authors describe the Hadley cells in Figure 4 "narrower" and "wider". Is it in the vertical direction ? Because they look in both cases 30 degrees wide ...

---

## Referee Comment (RC3) · Anonymous Referee #3 · 16 Nov 2017

Summary:

This paper explores a climate bifurcation for dry/desert planets, including implicit water cycling through the assumption of a sub-surface aquifer. This work represents a step forward from the classic work by Abe et al. 2011. I feel that this paper is interesting and deserves to be published. I suggest that the authors try to do more to explain why this transition happens, in terms of the cloud albedo and water vapor greenhouse forcing.

General Comments:

1) "It should be noted that the present paper is mainly descriptive in nature and is not meant to explain the mechanisms for the emergence of the two climate states and the way the transition between them happens. This is still under investigation."

[Figure]

You should explain the mechanisms for the emergence of the two climate states in this paper. This should be included in this paper to make it complete. I think you can do so by examining the changes in cloud albedo and water vapor greenhouse effect.

2) In your simulations, water is in theory made available everywhere on the planet via diffusion of water from the subsurface water-table into the atmosphere. How is the temperature, clouds, and water vapor in your terraplanet simulations different from an aquaplanet case, given an equal albedo? It would be useful to run 1 aquaplanet simulation with albedo = 0.07 for comparative purposes. (Note that 0.07 is fine for a typical ocean albedo).

Specific Comments:

1) I may suggest changing the title from "...Earth-like terra planet" to "...Earth-like Dune planet", or "...water-limited terrestrial planets." The term "terra-planet" isn't a common term in the literature. Also, the adjective "Earth-like" connotes a planet where surface liquid water is freely available. These dry worlds are perhaps more so Mars-like in description. OR, due to the ambiguity of the term "terra-planet", you should explicitly state somewhere early in the Abstract, perhaps the in the first sentence, that you are dealing with water-limited land planets.

2) Page 2, line 5 "Second, terra-planets with optically thin atmospheres (like present day Earth's atmosphere) can maintain their inner edge of the habitable zone much closer to their parent star compared to aqua-planets due to their higher surface albedo (Abe et al., 2011; Zsom et al., 2013)."

While it is true that dry planets will have a higher albedo than ocean planets, this is just one piece of the puzzle, and I don't think the dominant one. One of the primary reasons that dry planets can maintain habitability at higher stellar fluxes is due to the lack of available water vapor. With less water in the system, the water-vapor greenhouse feedback is severely muted and thus there is significantly less greenhouse warming of the climate system. This is true regardless of surface albedo.

3) Page 3, line 5. Can you give the model horizontal resolution in either degrees lon X degrees lat, or in number lon grids X number of lat grids?

4) what does "Leaf Area Index (LAI) =3" mean? Is your surface vegetated or bare soil? How might the vegetation type affect your results?

5) "overland water recycling" You are not really considering overland water recycling. You are considering a ubiquitous sub-surface aquifer that sources water into the model. Thus you are considering "subsurface" water recycling.

6) "$\alpha$ varying from 0.07 to 0.24" what kind of land surfaces are these appropriate for? igneous rock? desert sand?

7) Page 3, line 32. While the surface temperatures reach equilibrium quickly for dry planets, what about the atmosphere-land water balance? It is my understanding that in some models it can take a considerable amount of model time for water to permeate out of the sub-surface aquifer.

8) It would be interesting to see a figure similar to figure 5, but showing the integrated water vapor column amount, the integrated cloud water amount, and the integrated cloud fraction.

9) It would be interesting to see a additional panels on figure 7 showing cloud and TOA albedo, and also the greenhouse effect. You can estimate the greenhouse effect as G=sigma*Ts^4 - OLR

---

## Author Comment (AC1) · 21 Dec 2017

**Response to Referee #1 comments:**

**Overview:** The authors run a sophisticated global climate model (GCM) with a surface that is covered by a sophisticated land model. They assume a globally uniform ground water table, representing efficient surface water transport. Interestingly, they find bistability between two climate states, a hot and dry (HD) state and a cold and wet (CW) state. I have never seen anything like this before and it is definitely worth publishing. I think the authors should do a bit more work to understand why the model produces this behavior, then the paper will be ready to be published. I have some comments that hopefully will help with this.

*We thank our first referee Dr. Dorian Abbot for the very immediate yet a detailed review of our manuscript. Please find below our responses to his comments.*

**Comments:**
**Referee #1 comment - Mechanism:** In the final paragraph of the introduction the authors restrict the scope of the paper so that it doesn't include investigating the mechanism of bi-stability. I think at least some investigation of the mechanism is necessary so that this paper stands as an independent work that doesn't rely on future work and to convince the reader that the observed behavior isn't simply due to some bug in the land surface scheme.

*Authors' response: We do understand that including the mechanisms of bi-stability in the same paper would give a more complete picture to the reader. However, including the mechanisms explaining the emergence of the bi-stability would drastically increase the length of the paper. A complete paper on the mechanisms of bi-stability is already in preparation. If the Editor feels it is necessary, we could think about submitting our forthcoming paper on the mechanisms of bi-stability as a part two of the present paper to ESD as well so that together part one and part two papers could stand as an independent work.*

**Referee #1 comment:** The way I would approach this is to first run the model in "swamp" configuration: with a global mixed layer ocean of 1 m depth and zero ocean heat transport. Presumably you will reproduce a roughly Earth-like climate. Then I would turn on the land surface model, but choose schemes and parameters to match the swamp configuration as closely as possible. Then slowly turn more schemes on or change parameters until you get bistability. This should allow you to identify the specific physical parameterization that allows for bistability. It must be related to evapotranspiration, since that's the key difference between a swamp surface and a real land surface.

*Authors' response: We thank Referee #1 for suggesting an approach to better understand the mechanism behind the bi-stability. Incidentally, we already performed simulations of the type proposed by the Referee #1 and this was indeed how we found the bi-stability that we describe in our paper. Our model in the "Swamp configuration" is able to successfully reproduce the climate of an "Aqua-planet" configuration. Starting from this setup, we sequentially changed different parameters and schemes in our land surface model (Table 1) and we clearly saw that when the ground water table was lowered, the bi-stability emerged. In Table 1 we list our sequence of simulations and in Fig. 1 we show the resulting zonal structure for temperature and precipitation. With this additional information, we hope it is convincing enough for the reader to believe that the bi-stability is not due to some bug in the model's land surface scheme. We will include this information in the Appendix of the manuscript to demonstrate that indeed the restricted atmospheric access to water causes the appearance of the bi-stability.*

**Referee #1 comment – Surface energy balance:** Surface energy balance is important for global-mean evaporation, and therefore precipitation. All else equal, more absorbed shortwave at the surface should mean more evaporation and more precipitation. That's why it seems odd that the planetary albedo should be lower in the HD state, but the global-mean precipitation should also be so much lower.

*Authors' response: Referee #1's statement "more absorbed shortwave at the surface should mean more evaporation and more precipitation" – is only valid for an Earth-like planet with large oceans. However, in our HD state, even though the tropics receive huge amounts of net radiation at the surface, the uppermost soil layers in the tropics dry out quickly. Dry uppermost soil layers imply small evapotranspiration leading to no precipitation which in turn leads to even less evapotranspiration. This self stabilizing mechanism maintains the HD state.*

**Referee#1 comment:** One possibility is that much more of the surface heat is lost through sensible rather than latent heat in the HD state. Another is that the planetary shortwave absorption is not a good proxy for surface shortwave absorption because of differential atmospheric absorption of radiation in the two cases. I think it's worth calculating the terms in the surface energy balance (both zonal mean and global mean) and using this to explain why the HD state seems to be able to absorb so much more shortwave yet have such a low evaporation. If you can explain this, it might also help your investigation into the mechanism for bistability.

*Authors' response: We agree with the Referee #1 about his first suggestion that much more of the surface heat is lost through sensible rather than through latent heat in the HD state (see Fig. 2 below). Additionally, most of the heat is lost from the surface by terrestrial radiation in HD state. Upon re-submission we would include a brief discussion about the surface energy balance in section 3 of our revised manuscript and add a figure in the appendix (Fig. 2 here).*

**Referee #1 comment:** Also, I wonder if this could be connected to why bistability is lost for Earth-like obliquity. It would be really great if you could explain why increasing obliquity disrupts the bistability, and maybe analyzing the surface energy balance would help.

*Authors' response: We have so far not analyzed the reason why bi-stability is lost for Earth-like obliquity in detail and it is one of our plans for the future. But we speculate that for Earth-like obliquity, the bi-stability is lost due to the seasonal migration of the rain bands towards the lower latitudes. The seasonal migration of rain bands facilitates seasonal rain in the dry tropics. Since soil moisture has a memory lasting for several weeks to months this causes the soils in the tropics to remain wet even during the dry season. Thus there is always soil moisture in the originally dry tropics to allow for continuous evapotranspiration and precipitation. And this destroys the HD state so that the planet is always self stabilized in the CW state. We will include this speculation in the section about seasonality in our revised manuscript.*

**Referee #1 comment – Vertical Temperature Structure:** One thing I was wondering about is the vertical temperature structure in the two states, since convection is probably important for the bistability. I think it would be worth plotting and thinking about this.

*Authors' response: We have attached the figure for vertical temperature profile including potential temperature in the tropics for the two terra-planet states (see Fig. 3 below). But vertical temperature structure does not explain much about the bi-stability, the restricted atmospheric access to water causes the appearance of the bi-stability*

**Referee #1 comment –** A related point is that near the outer edge of the habitable zone we expect very high CO2 levels, probably at least 1 bar. A large radiative cooling in the atmosphere could strongly affect the vertical temperature structure. It would be interesting to do some test runs at very high CO2 and see if the HD state can persist under these conditions, since the authors connect the HD state with the outer edge of the habitable zone.

*Authors' response: We thank the Referee #1 for his interesting suggestion. Unfortunately, we cannot run such simulations as our model parameterizations are not suited for very high $CO_2$ concentrations and hence it is beyond the scope of the present paper.*

| Simulations | Background soil albedo | Surface roughness | Heat capacity | Snow albedo | Water reservoir depth |
|---|---|---|---|---|---|
| Aqua-planet | 0.07 | Ocean | Ocean | 0.07 | 50 m slab ocean, no heat transport |
| Swamp1 | 0.07 | Ocean | Ocean | 0.07 | Constant ground water table at a depth of 0.3 m |
| Swamp2 | 0.07 | Land | Land | 0.07 | Constant ground water table at a depth of 0.3 m |
| HD | 0.07-0.14 | Land | Land | Dynamic (0.4 -0.8) | Constant ground water table at a depth of 1.2 m |
| CW | 0.15-0.24 | Land | Land | Dynamic (0.4 -0.8) | Constant ground water table at a depth of 1.2 m |

Table 1. Summary of simulations performed to illustrate the procedure followed by which we found bi-stability on our terra-planet.

[Figure]

Figure 1: (a) Time series of global mean surface temperature (°C) and (b) annual mean meridional profile of precipitation (mm day$^{-1}$) for different simulations performed to illustrate the procedure by which we found bi-stability on our terra-planet.

[Figure]

**Figure 2: Annual mean meridional profiles of Net shortwave radiation at surface (SW), latent heat flux (Latent) and sensible heat flux (Sensible) for the two terra-planet states: HD ($\alpha$ = 0.14) and CW ($\alpha$ = 0.15) in the simulations Z14 and Z15 averaged over a period of ten years.**

[Figure]

**Figure 3: Vertical profiles of Temperature and potential temperature in the tropics for the two terra-planet states in the simulations Z14 (HD state) and Z15 (CW state) averaged over a period of ten years.**

---

## Author Comment (AC2) · 21 Dec 2017

**Response to Referee #2 comments:**

**General comments**
**Referee #2 comment:** In this paper, the authors modelled the climate of a terra planet (or land planet), a planet with a global land surface, without ocean. The particularity of the present study is the presence of an unlimited underground water layer providing water at all latitude but behaving differently as an ocean. The authors perform various simulations, changing obliquity, snow albedo and a test with a different convection scheme.

Depending on the value of the surface albedo, two different stable climate states appear. A hot and dry state and a cold and wet state. This latter is a novelty, due to the addition of the underground water reservoir and is an interesting result. However, as the main novelty is the presence of an underground reservoir, more discussion would be useful. For instance the authors tested only a fixed depth of 1.2m to mimic a "recycling" of water from higher to lower latitudes. This setup is somewhat artificial and a justification of the current assumptions as well as the potential effect of varying them would be welcomed.

*Authors' response: We thank the referee #2 for the comment. Our main idea behind this study was to explore how water recycling can shape the climate of water-limited planets. In the traditional land planet studies with limited water inventories, all the water on the planet is exported to the higher latitudes by atmospheric circulation where it is permanently piled up as snow on the icecaps and is no longer a part of the hydrological cycle and has no effect on the climate. However, in reality there is a limit on how large an ice cap can grow and beyond a particular threshold, parts of the ice cap can break off and flow towards warmer regions due to gravity and lead to liquid water formation. Also, as the thickness of the ice cap increases, geothermal flux at the bottom of the ice caps can cause melting and liquid water formation. This water then flows back to the lower latitudes through river flows or if the surface is porous like our present-day Earth the water can percolate into the ground and refill some kind of underground reservoir. Thus, it again becomes a part of the hydrological cycle and therefore can affect the climate in the lower latitudes. In our study, we mimic such a water recycling from higher to lower latitudes by means of a homogenous global subsurface reservoir. Indeed, referee #2 is correct concerning our assumption of a homogenous recycling over the whole globe to be artificial. In reality, recycling may occur at different speeds and might be less or more effective than what we consider in our study. Based on the speed of recycling, the water level of this subsurface reservoir would vary. In our study, the choice of the fixed ground water table depth of 1.2 m was not random and we performed a sequence of simulations starting from a swamp simulation and changed different parameters (including the depth of the ground water table) and various schemes in our land surface model (Table 1; Fig. 1) until we arrived at the bi-stability for a ground water table depth of 1.2 m. Overall, our study should be considered as a first attempt towards understanding how water recycling can shape the climate of land planets with reasonable assumptions. We will include all the information regarding the sequence of simulations performed to arrive at the bi-stability in the appendix of the revised manuscript.*

**Specific comments**
**Referee #2 comment:** lines 29-30 of page 4. The authors describe the Hadley cells in Figure 4 "narrower" and "wider". Is it in the vertical direction? Because they look in both cases 30 degrees wide ...

*Authors' response: The reviewer is indeed correct that the Hadley cells in both the terra-planet states look very similar and have the same width close to the surface. But with height the Hadley cell in the CW state gets slightly narrower compared to that in the HD state (when measured around 500hPa). We will rewrite the sentences related to width of the Hadley cell in the revised manuscript to make our point clear.*

| Simulations | Background soil albedo | Surface roughness | Heat capacity | Snow albedo | Water reservoir depth from the surface |
|---|---|---|---|---|---|
| Aqua-planet | 0.07 | Ocean | Ocean | 0.07 | 50 m slab ocean, no heat transport |
| Swamp1 | 0.07 | Ocean | Ocean | 0.07 | Constant ground water table at a depth of 0.3 m |
| Swamp2 | 0.07 | Land | Land | 0.07 | Constant ground water table at a depth of 0.3 m |
| HD | 0.07-0.14 | Land | Land | Dynamic (0.4 -0.8) | Constant ground water table at a depth of 1.2 m |
| CW | 0.15-0.24 | Land | Land | Dynamic (0.4 -0.8) | Constant ground water table at a depth of 1.2 m |

**Table 1. Summary of simulations performed to illustrate the procedure by which we found bi-stability on our terra-planet.**

[Figure]

**Figure 1: (a) Time series of global mean surface temperature (°C) and (b) annual mean meridional profile of precipitation (mm day$^{-1}$) for different simulations performed to illustrate the procedure by which we found bi-stability on our terra-planet.**

---

## Author Comment (AC3) · 21 Dec 2017

**Response to Referee #3 comments:**

**Summary:**
This paper explores a climate bifurcation for dry/desert planets, including implicit water cycling through the assumption of a sub-surface aquifer. This work represents a step forward from the classic work by Abe et al. 2011. I feel that this paper is interesting and deserves to be published. I suggest that the authors try to do more to explain why this transition happens, in terms of the cloud albedo and water vapor greenhouse forcing.

*We thank our referee for very interesting comments and suggestions on our manuscript. Please find below our responses to the referee#3 comments.*

**General comments:**
**Referee #3 comment:** "It should be noted that the present paper is mainly descriptive in nature and is not meant to explain the mechanisms for the emergence of the two climate states and the way the transition between them happens. This is still under investigation." You should explain the mechanisms for the emergence of the two climate states in this paper. This should be included in this paper to make it complete. I think you can do so by examining the changes in cloud albedo and water vapor greenhouse effect.

*Authors' response: We thank the Referee#3 for suggesting us to look at the role of clouds and water vapor greenhouse forcing to explain the emergence of two drastically different states. Indeed, it is true that the two terra-planet climate states display considerably different patterns of cloud cover and water vapor in the lower latitudes (Fig. 1). In the HD state, the cloud cover in the low latitudes is exclusively composed of high level clouds with very low liquid water content (Fig. 2). The reason being, the high surface temperatures in the lower latitudes in the HD state raise the water vapor saturation limit of the atmosphere and the height at which condensation and cloud formation occurs. High clouds are more transparent to shortwave radiation, at the same time they reduce outgoing longwave radiation and thereby keep the planet hot and stabilize the HD state. With an increase in background soil albedo beyond the bifurcation threshold, the water vapor saturation limit of the atmosphere and the height at which the clouds can form is lowered due to the lowering of surface temperature, thus leading to an increase in low level cloud cover. Increase in low level cloud cover increases the planetary albedo and keeps the planet cool and self stabilized in the CW state. Overall, the self stabilizing cloud albedo feedback does play a role in the transition from the HD to the CW state (we briefly mentioned this in Sections 6 and 8 in the original manuscript). However, a complex sequence of steps is involved in the formation of low level clouds before the self stabilization of the planet in the CW state by the cloud albedo feedback. Including all these complex mechanisms would drastically increase the length of the paper and hence we prefer to present the explanations regarding the mechanisms involved in the transition in a separate paper (already in preparation). If the Editor feels it is necessary, we could think about submitting our forthcoming paper on the mechanisms of bi-stability as a part two of the present paper to ESD as well so that together part one and part two papers could stand as an independent work.*

**Referee #3 comment:** In your simulations, water is in theory made available everywhere on the planet via diffusion of water from the subsurface water-table into the atmosphere. How is the temperature, clouds, and water vapor in your terraplanet simulations different from an aquaplanet case, given an equal albedo? It would be useful to run 1 aquaplanet simulation with albedo = 0.07 for comparative purposes. (Note that 0.07 is fine for a typical ocean albedo).

*Authors' response: This is part of what we already did: We performed a series of simulations starting from an aqua-planet setup to arrive at the bi-stability (please see Fig. 3 and Table 1 below). In Fig. 4 below you find the requested plots for annual mean meridional profiles of (a) surface temperature, (b) vertically integrated water vapor, (c) vertically integrated cloud cover for an aqua-planet simulation with albedo = 0.07 and terra-planet simulations with the same background soil albedo value (0.07) but initialized differently: from a warmer state (HD – 0.07 simulation) and from a colder state (CW - 0.07*

*simulation).*

*The aqua-planet climate resembles more closely to the CW - 0.07 simulation and is about 22K colder than the HD-0.07 simulation. However, the atmosphere in the terra-planet simulation (HD – 0.07) can hold a similar amount of water vapor as in the aqua-planet simulation due to higher temperatures in HD-0.07 (Fig. 4b). The cloud cover at the lower latitudes in terra-planet simulation (HD – 0.07) is lower compared to that of the aqua-planet simulation (Fig. 4c) and is mainly comprised of high level clouds whereas in the aqua-planet case (also for CW-0.07 simulation), it is dominated by low level clouds.*

**Specific comments:**

**Referee #3 comment:** I may suggest changing the title from ". . .Earth-like terra planet" to ". . .Earth-like Dune planet", or ". . .water-limited terrestrial planets." The term "terra-planet" isn't a common term in the literature. Also, the adjective "Earth-like" connotes a planet where surface liquid water is freely available. These dry worlds are perhaps more so Mars-like in description. OR, due to the ambiguity of the term "terra-planet", you should explicitly state somewhere early in the Abstract, perhaps the in the first sentence, that you are dealing with water-limited land planets.

*Authors' response: Our main idea behind naming the planet as terra is to contrast it from the well known aqua-planet. The word "terra" means "land" in Latin. To avoid ambiguity regarding the term "Earth-like", we shall in our revised manuscript explicitly define "Earth-like terra-planet" as a water limited terrestrial planet in the first sentence of the abstract.*

**Referee #3 comment:** Page 2, line 5 "Second, terra-planets with optically thin atmospheres (like present day Earth's atmosphere) can maintain their inner edge of the habitable zone much closer to their parent star compared to aqua-planets due to their higher surface albedo (Abe et al., 2011; Zsom et al., 2013)." While it is true that dry planets will have a higher albedo than ocean planets, this is just one piece of the puzzle, and I don't think the dominant one. One of the primary reasons that dry planets can maintain habitability at higher stellar fluxes is due to the lack of available water vapor. With less water in the system, the water-vapor greenhouse feedback is severely muted and thus there is significantly less greenhouse warming of the climate system. This is true regardless of surface albedo.

*Authors' response: We thank the referee#3 for pointing out the primary reason why dry planets can maintain habitability even at higher stellar fluxes. We will modify the text as suggested by the referee#3 in our revised manuscript.*

**Referee #3 comment:** Page 3, line 5. Can you give the model horizontal resolution in either degrees lon X degrees lat, or in number lon grids X number of lat grids?

*Authors' response: The model has a horizontal resolution of R2B04 equivalent to a resolution of an evenly distributed rectangular grid of about ~ 160 Km. We would modify the text in the revised manuscript to make it more clear for the reader.*

**Referee #3 comment:** what does "Leaf Area Index (LAI) =3" mean? Is your surface vegetated or bare soil? How might the vegetation type affect your results?

*Authors' response: Yes, there is something that one could call vegetation, because our model includes besides bare soil evaporation also a transpiration pathway for soil water to reach the atmosphere. The value of LAI controls how much surface in a grid cell participates in transpiration, while the rest exhibits bare soil evaporation. But we consider the value of the LAI here only as a means to parameterize the atmospheric access to water, like other hydrological parameters of the model, e.g. soil porosity, hydraulic conductivity, or depth of the 'subsurface reservoir'. Therefore, and because other aspects of vegetation like plant growth or phenology are missing, we do not use the word 'vegetation' in our paper. We would rewrite the the text regarding the LAI in the revised manuscript to make it more clear for the reader.*

**Referee #3 comment:** "overland water recycling" You are not really considering overland water recycling. You are considering a ubiquitous sub-surface aquifer that sources water into the model. Thus you are considering "subsurface" water recycling.

*Authors' response: We agree with the Referee#3's comment regarding the term "overland water recycling". However, it also not just "subsurface" water recycling but imagine it as a combination of recycling happening via ice/river flows over the land and, subsurface flows. We will try to find a more concise and appropriate term which represents both these flows for our revised manuscript.*

**Referee #3 comment:** "_ varying from 0.07 to 0.24" what kind of land surfaces are these appropriate for? igneous rock? desert sand?

*Authors' response: Vegetated surfaces, but also deserts with mainly rocky surfaces have albedo values within this range. Sandy deserts typically have higher albedo values.*

**Referee #3 comment:** Page 3, line 32. While the surface temperatures reach equilibrium quickly for dry planets, what about the atmosphere-land water balance? It is my understanding that in some models it can take a considerable amount of model time for water to permeate out of the sub-surface aquifer.

*Authors' response: Referee#3's remark "it can take a considerable amount of model time for water to permeate out of the sub-surface aquifer" is indeed valid in the case of a geological reservoir on a real planet where recycling does have long internal timescales and the equilibration of surface temperature may require a considerably longer time than spanned by our simulations. However, the sub-surface reservoir that we consider in our study just refers to the bottom two layers of the soil. The total soil depth in our study is about 10 meters of which the layers below a depth of 1.2 meters are almost completely filled with water homogenously all over the globe and should not be misunderstood as a geological reservoir. Only the top three layers of the soil are dynamic which take around three years to equilibrate.*

**Referee #3 comment:** It would be interesting to see a figure similar to figure 5, but showing the integrated water vapor column amount, the integrated cloud water amount, and the integrated cloud fraction.

*Authors' response: As requested, we show in figure 5 below the vertically integrated water vapor column amount, vertically integrated cloud water amount, and vertically integrated cloud fraction for different stages of the transition from the HD state to the CW state similar to figure 5 in the original manuscript.*

*In the HD state, the lower latitudes are mainly covered by high cloud cover (Fig. 2) with very low liquid water content and huge amount of water vapor in the atmosphere.*

*Stage 1: Once the albedo is increased beyond the threshold, precipitation clusters start to appear close to the equator, more rain reaches the surface filling up the dry uppermost soil layers in the lower latitudes. With increased soil moisture availability, evapotranspiration increases resulting in a transient increase in atmospheric water vapor content and cloud liquid water content in the lower latitudes (Fig. 5).*

*Stage 2 & 3: As the temperature drops further in the later stages, water vapor saturation limit and the height at which clouds can form is lowered leading to an increase in low level cloud cover with high liquid water content at the lower latitudes. Increase in low level cloud cover increases the planetary albedo and results in the rapid cooling of the surface and a decrease in atmospheric water vapor content and eventual stabilization of the CW state.*

**Referee #3 comment:** It would be interesting to see additional panels on figure 7 showing cloud and TOA albedo, and also the greenhouse effect. You can estimate the greenhouse effect as G=sigma*Ts^4 – OLR

*Authors' response: As requested, we show in figure 6 below cloud albedo, TOA albedo, and greenhouse effect as a function of background soil albedo similar to Figure 7 in the original manuscript. We notice that for an increase in background soil albedo, the planetary albedo increases due to an increase in low level cloud cover at lower latitudes.*

*Also, at a higher background soil albedo, the greenhouse effect is much lower compared to that for lower background soil albedo due to decrease in atmospheric water vapor greenhouse warming (Fig. 6c).*

[Figure]

**Figure 1: Annual mean meridional profiles of precipitable water (top panel) and cloud cover fraction (bottom panel) for the two terra-planet states: HD ($\alpha$ = 0.14) and CW ($\alpha$ = 0.15) in the simulations Z14 and Z15 averaged over a period of ten years.**

[Figure]

**Figure 2: Vertical profiles of cloud liquid water content (g kg⁻¹) and cloud area fraction for the two terra-planet states in the simulations Z14 and Z15 averaged over a period of ten years.**

| Simulations | Background soil albedo | Surface roughness | Heat capacity | Snow albedo | Water reservoir depth from the surface |
|---|---|---|---|---|---|
| **Aqua-planet** | 0.07 | Ocean | Ocean | 0.07 | 50 m slab ocean, no heat transport |
| **Swamp1** | 0.07 | Ocean | Ocean | 0.07 | Constant ground water table at a depth of 0.3 m |
| **Swamp2** | 0.07 | Land | Land | 0.07 | Constant ground water table at a depth of 0.3 m |
| **HD** | 0.07-0.14 | Land | Land | Dynamic (0.4 -0.8) | Constant ground water table at a depth of 1.2 m |
| **CW** | 0.15-0.24 | Land | Land | Dynamic (0.4 -0.8) | Constant ground water table at a depth of 1.2 m |

**Table 1. Summary of simulations performed to illustrate the procedure by which we found bi-stability on our terra-planet.**

[Figure]

**Figure 3: (a) Time series of global mean surface temperature (°C) and (b) annual mean meridional profile of precipitation (mm day[-1]) for different simulations performed to illustrate the procedure by which we found bi-stability on our terra-planet.**

[Figure]

**Figure 4: Annual mean meridional profiles of (a) surface temperature (°C), (b) vertical integrated water vapour (mm), (c) cloud cover fraction for an aqua-planet simulation with albedo = 0.07 and terra-planet simulations with the same background soil albedo value (0.07) but initialized differently: from a warmer state (HD – 0.07 simulation) and from a colder state (CW - 0.07 simulation).**

[Figure]

**Figure 5: Annual mean vertically integrated water vapor (left panels), vertically integrated cloud water (centre panels) and cloud cover cover fraction (right panels) for different stages of the transition (TRANS simulation) from the HD state to the CW state similar to the Fig. 5 in the original manuscript.**

[Figure]

**Figure 6:  TOA albedo (a), cloud albedo (b) and Greenhouse effect (c) plotted as a function of background soil albedo (α). Path 1 denotes the spontaneous transition from the HD to the CW state when increasing background soil albedo and paths 2, 3 and 4 denote the reverse state development upon stepwise lowering of background albedo starting from the threshold value (α = 0.15) to α = 0.**

---

## Referee Report (RR1)

The paper suggests that there are two drastically different climate states on an Earth-like terra-planet with limited surface water. This is a very interesting paper and it is worth publishing on ESD. I carefully read the manuscript and the comments from other three reviewers as well as the authors' responses. The paper is clear and easy to understand for readers, although the mechanism for the bi-stability between the cold wet (CW) state and the hot dry (HD) state is still unclear. The authors have done many analyses, including precipitation, precipitable water, soil moisture, clouds, convection, the Hadley cells, poleward energy transports, surface/planetary albedos, etc. These analyses make the authors speculate that the bi-stability is from a combination of cloud and hydrological feedbacks. How can cloud trigger the bi-stability? If possible, the story should not end here.

As well known, ice/snow albedo feedback causes two different climate states, a modern Earth climate state (or even hotter) and a snowball Earth state. The mechanism is clear, i.e., the step function of surface albedo, from 0.4-0.9 of ice/snow when the surface temperature is below the freezing point to < 0.1 of ice-free seawater when the surface temperature is above $0^{o}C$. For the terra-planet discussed in the manuscript, the cause is likely the clouds, as addressed in the manuscript, but how clouds produce the bi-stability is unclear. Is there any step function in the cloud parameterization of the climate model used in this work?

From Figures 2, 8, 9 & B1, the main difference in cloud coverage between the CW and HD states is in the subtropics, where the atmosphere should be dry and relative humidity is low. In the deep tropics and at the Polar Regions, the difference in cloud cover is very small. So, more analyses should focus on the subtropical clouds. For example, "Cloud cover is calculated based on relative humidity" (page 3, line 10 of the manuscript). In the model, is the cloud coverage a step function of relative humidity? The authors could check this and meanwhile some analyses on atmospheric relative humidity could be added in the manuscript if it is important.

Jun Yang

Dept. of Atmospheric and Oceanic Sciences,

Peking University, Beijing, China.

April 10, 2018

---

## Author Response (AR2)

Dear Dr.Valerio Lucarini,

We thank you and the reviewers for the positive response. Based on the comments from you and the two reviewers we have now prepared a revised manuscript. Please find below point by point responses to reviewers' comments and the revised manuscript with all the changes highlighted in blue.

With best regards,

Sirisha Kalidindi, Christian H. Reick, Thomas Raddatz and Martin Claussen

**Point-by-point response to reviewers' comments**

**Response to Reviewer #1 comments:**

I'm happy with the responses the authors have made to the reviews. I would prefer to see the mechanism laid out in full in this paper, but it's up to the authors to decide how to get their work out. One significant issue remains: I didn't see any actual revisions to the paper, just comments in the responses to the reviews. Am I missing something? I'm not sure if I'm just confused with this online journal. I think some revisions to the paper according to the reviews really should be made, and I'm happy to sign off on the paper once they are.

*Authors' response: We thank Dr. Dorian Abbot for the positive feedback to our responses. We appreciate his view that it is up to us to decide whether to include/exclude the explanations for the mechanism for the bi-stability.*

**Response to Referee #2 comments:**

**Comment 1:** The paper suggests that there are two drastically different climate states on an Earthlike terra-planet with limited surface water. This is a very interesting paper and it is worth publishing on ESD. I carefully read the manuscript and the comments from other three reviewers as well as the authors' responses. The paper is clear and easy to understand for readers, although the mechanism for the bi-stability between the cold wet (CW) state and the hot dry (HD) state is still unclear. The authors have done many analyses, including precipitation, precipitable water, soil moisture, clouds, convection, the Hadley cells, poleward energy transports, surface/planetary albedos, etc. These analyses make the authors speculate that the bi-stability is from a combination of cloud and hydrological feedbacks. How can cloud trigger the bistability? If possible, the story should not end here.

*Authors' response: We thank Dr. Jun Yang for the positive comments on our manuscript. Regarding the mechanisms responsible for the bi-stability, we speculate that a combination of cloud and hydrological feedbacks to be responsible for the bi-stability because our analysis shows that these feedbacks work together (although in a different way) in stabilizing the two climate states (described shortly in sections 3.3 and 3.4 of the revised manuscript). Coming to the question of the role of clouds in the initiation of the bifurcation, our preliminary analysis suggests that the bifurcation is trigged by a re-organization of the hydrological cycle in the low-latitudes and that the clouds have only an amplifying effect towards the further stabilization of the planet in the CW state by increasing the planetary albedo (page 6, lines 20- 21 in the revised manuscript). We have made this clear now for the reader by rewriting the text in the discussion section of the revised manuscript (page 9, lines 31-32).*

**Comment 2:** As well known, ice/snow albedo feedback causes two different climate states, a modern Earth climate state (or even hotter) and a snowball Earth state. The mechanism is clear, i.e., the step function of surface albedo, from

0.4-0.9 of ice/snow when the surface temperature is below the freezing point to < 0.1 of ice-free seawater when the surface temperature is above 0°C. For the terra-planet discussed in the manuscript, the cause is likely the clouds, as addressed in the manuscript, but how clouds produce the bi-stability is unclear. Is there any step function in the cloud parameterization of the climate model used in this work?

***Authors' response:*** *We thank the reviewer for the interesting comment. Firstly, the cloud cover parameterization in our model is not defined as a step function. Secondly, our preliminary analysis suggests that for the terra-planet, the mechanism triggering the bifurcation is much more complex than just a simple step function of ice-albedo as for the Snowball Earth bifurcation. The bifurcation involves a complex sequence of atmospheric and hydrological re-organizations before the self stabilization of the planet in the CW state. We are still investigating this and the outcome will be published as a separate paper.*

**Comment 3:** From Figures 2, 8, 9 & B1, the main difference in cloud coverage between the CW and HD states is in the subtropics, where the atmosphere should be dry and relative humidity is low. In the deep tropics and at the Polar Regions, the difference in cloud cover is very small. So, more analyses should focus on the subtropical clouds.

***Authors' response:*** *We thank the reviewer for this comment. Indeed, the cloud cover in the sub-tropics in the two states is very different. However, for the bifurcation, the mechanisms occurring at the equator are more relevant since the bifurcation begins in the tropics. Hence, in the manuscript we do not include any additional information regarding the subtropical clouds.*

**Comment 4:** For example, "Cloud cover is calculated based on relative humidity" (page 3, line 10 of the manuscript). In the model, is the cloud coverage a step function of relative humidity? The authors could check this and meanwhile some analyses on atmospheric relative humidity could be added in the manuscript if it is important.

***Authors' response:*** *Cloud cover in our model is not defined as a step function of relative humidity. Our preliminary analysis suggests that the transition between the two climate states on a terra-planet is mostly determined by a re-organization of the hydrological cycle in the low-latitudes and cloud feedbacks only provide an amplifying effect.*

[revised manuscript text omitted]